# Test-Time Adaptation for Combating Missing Modalities in Egocentric Videos

**Merey Ramazanova, Alejandro Pardo, Bernard Ghanem, Motasem Alfarra**
Center of Excellence in Generative AI, KAUST, Saudi Arabia
`firstname.lastname@kaust.edu.sa`

## Abstract

Understanding videos that contain multiple modalities is crucial, especially in egocentric videos, where combining various sensory inputs significantly improves tasks like action recognition and moment localization. However, real-world applications often face challenges with incomplete modalities due to privacy concerns, efficiency needs, or hardware issues. Current methods, while effective, often necessitate retraining the model entirely to handle missing modalities, making them computationally intensive, particularly with large training datasets. In this study, we propose a novel approach to address this issue at test time without requiring retraining. We frame the problem as a test-time adaptation task, where the model adjusts to the available unlabeled data at test time. Our method, MiDl (Mutual information with self-Distillation), encourages the model to be insensitive to the specific modality source present during testing by minimizing the mutual information between the prediction and the available modality. Additionally, we incorporate self-distillation to maintain the model's original performance when both modalities are available. MiDl represents the first self-supervised, online solution for handling missing modalities exclusively at test time. Through experiments with various pretrained models and datasets, MiDl demonstrates substantial performance improvement without the need for retraining. Our code is available in this repo.

## 1 Introduction

Understanding multimodal data has emerged as a pivotal challenge in various domains, including foundational model construction (Radford et al., 2021), emotion recognition (Lee et al., 2019), and analysis of egocentric videos (Grauman et al., 2023; 2022; Damen et al., 2018) for tasks like recognition (Kazakos et al., 2021; Nagrani et al., 2021; Xiao et al., 2020) and localization (Ramazanova et al., 2023; Wang et al., 2023a). Recent efforts focused on crafting models harnessing data diverse modalities showing performance gains in multiple tasks (Lin et al., 2022). However, a critical limitation arises: these models presuppose complete modality availability at test-time, which diverges from real-world scenarios where data modalities may be incomplete (Ma et al., 2022; Ramazanova et al., 2024). For instance, in real-time prediction using wearable devices, portions of recordings might be redacted to safeguard privacy, or cost constraints may necessitate using cheaper modalities like audio or IMU (Grauman et al., 2022; 2023). Consequently, models designed under this assumption demonstrate significant performance degradation in the face of missing modalities, sometimes even underperforming unimodal counterparts trained with a single modality (Ramazanova et al., 2024).

The challenge of addressing missing modalities gained significant attention from researchers recently, where various strategies have been proposed to mitigate this issue (Neverova et al., 2015; Ma et al., 2021; 2022; Ramazanova et al., 2024). Some works have explored architectural modifications to fuse information from different modalities effectively (Ma et al., 2022). Additionally, other approaches have focused on designing effective regularizers to boost model performance when confronted with missing modalities (Colombo et al., 2021). More recently, a promising direction has emerged, wherein transformer models have been augmented with learnable tokens during training (Lee et al., 2023; Ramazanova et al., 2024). These tokens serve to compensate for missing information at test time, significantly enhancing model robustness against missing modality (Ramazanova et al., 2024). Despite these advancements, a common drawback persists: all existing approaches necessitate expensive retraining of the multimodal model, rendering pretrained models obsolete. This poses a

Figure 1: **Test-Time Adaptation for missing modalities.** The concept of test-time adaptation in the presence of missing data modalities focuses on a system where a stream of multimodal data is input, potentially lacking one or more modalities. Without adaptation, the pretrained model $f_{\theta_0}$ may predict inaccurate labels due to incomplete data. With test-time adaptation, the model is dynamically adjusted using the adaptation method $g$, resulting in an adapted model $f_{\theta_t}$, designed to handle the missing modalities and improve over time. The graph on the right illustrates the performance of the non-adapted baseline (blue) vs. the model adapted with our proposed adaptation method MiDl (green) on Epic-Kitchens dataset. It shows the adaptation efficacy in maintaining higher performance levels despite the variability in modal-completeness, surpassing the unimodal performance (orange) for all missing rates.

substantial challenge, particularly in applications with extensive training data, where the retraining process is prohibitively expensive, making the aforementioned approaches impractical. Consequently, a fundamental question arises:

*Can we develop methods to address missing modalities at test time without imposing retraining requirements?*

In this work, we take the initial step of framing the missing modality problem as a test-time adaptation problem (Wang et al., 2020; Liang et al., 2020; Li et al., 2016). Specifically, we aim to establish a new approach wherein pretrained models undergo adaptation at test time to optimize their performance in the presence of missing modalities. Our formulation assumes a scenario where a continuous stream of unlabeled data is fed into the pretrained model during testing, with some instances missing certain modalities. The objective is to devise an adaptation algorithm capable of refining the model's predictions under missing modality in real-time settings (refer to Figure 1). Based on this formulation, we first evaluate existing methodologies from the test-time adaptation literature and demonstrate their limited efficacy in addressing this specific multimodal challenge. Subsequently, we introduce a novel test-time adaptation technique explicitly tailored to tackle the missing modality problem. Our method revolves around incentivizing the output of the pretrained model to remain *invariant* to the modality source present during testing. To achieve this, we propose minimizing the mutual information between the model's output and the modality type of the incoming unlabeled data at test time in a self-supervised manner. Moreover, to ensure the preservation of model performance under a complete modality setup, we integrate our approach with a self-distillation mechanism. Notably, our method, termed MiDl (Mutual information with self-Distillation minimization), is theoretically motivated and agnostic to the choice of pretrained model architecture, the dataset, and the specific type of missing modality encountered during testing.

We summarize **our contributions** in two-fold: **(1)** We redefine the missing modality problem as a test-time adaptation challenge, pioneering a novel approach where pretrained models are adapted at test-time to optimize performance in the face of missing modalities. We evaluate the effectiveness of the current adaptation methods under this challenging problem. **(2)** We introduce MiDl (Mutual information with self-Distillation minimization), a versatile test-time adaptation method explicitly designed to address the missing modality problem. MiDl ensures that model outputs remain invariant to modality sources during testing, enhancing robustness. It is agnostic to factors such as the pretrained model architecture, training dataset, and the specific type of missing modality, making it a comprehensive solution for diverse scenarios. When combined with pretrained models, MiDl achieves significant performance improvements, including a 6% gain on the Epic-Sounds dataset and an 11% gain on the Epic-Kitchens dataset.

## 2 RELATED WORK

**Missing Modalities in Multimodal Datasets.** Several works addressed the problem of missing modality in the multimodal datasets (Tsai et al., 2018; Ma et al., 2021; Zhao et al., 2021; Neverova

et al., 2015; Ma et al., 2022). Most methods addressing the missing modality problem assume full access to the source (training) data. Some works are based on the assumption that the training data is modal-complete, and the goal is to train a model robust to the missing inputs at test time (Ma et al., 2021; 2022). For example, Dai et al. (2024) investigate a strategy of randomly dropping video frames during training to improve the robustness of a multimodal system. Similarly, Lee et al. (2019) propose a method to train a network capable of generating audio features to handle missing modalities. Wang et al. (2023b) focus on a multimodal learning approach that models shared and specific features for classification and segmentation tasks. Other works tackle the modality distillation task, where the training data is multimodal, but only one modality is used at test time (Radevski et al., 2023; Garcia et al., 2019). Few works assume the modalities could be missing at train and test times and attempt to train a robust network (Lee et al., 2023; Ramazanova et al., 2024). In our work, we explore a more realistic scenario where we might not have access to the training data or network re-training is not feasible. We formulate this setup as a test-time adaptation problem, where the model is experiencing a distribution shift caused by the unavailability of some modalities at test time. Further, we propose the first test-time adaptation algorithm tailored to combat the missing modality challenge at test-time.

**Test-Time Adaptation.**    Test-time Adaptation (TTA) attempts to combat performance gaps that pretrained models suffer from when exposed to distribution shifts at test-time (Mancini et al., 2018; Kojima et al., 2022). This is usually attained through modifying the model's parameters (Liang et al., 2020) or its input (Gao et al., 2022) by using the incoming unlabeled data at test-time. TTA methods are practical, as they avoid assumptions on the training phase of a given model (Wang et al., 2020). The first of these approaches adjusts the statistics of the Batch Normalization (BN) layers (Li et al., 2016). This was followed by more powerful adaptation methods that involved self-supervised objective functions such as entropy minimization (Wang et al., 2020; Niu et al., 2022; Niu14 et al., 2023), information maximization (Liang et al., 2020), teacher-student approaches (Yuan et al., 2023), and leveraging auxiliary tasks (Alfarra et al., 2025). While such TTA methods made significant progress towards combating distribution shifts at test-time, they solely focused on simple covariate shifts such as changes in weather conditions or pixel illumination (Hendrycks & Dietterich, 2019). In this work, we extend the problem formulation of test-time adaptation to a very practical and realistic domain shift: missing modality. In particular, we adapt the stream setting of online test-time adaptation (Alfarra et al., 2023) to formulate the missing modality problem along with the corresponding evaluation protocol. Building on this novel view of the missing modality problem, we analyze the current state-of-the-art TTA methods where we show their limited impact. Further, we propose a novel TTA method tailored to combat the missing modality problem.

## 3    MISSING MODALITY AS TEST-TIME ADAPTATION

We first formulate the missing modality problem as a test-time adaptation problem in Section 3.1. We then outline the evaluation protocol for a given adaptation method in Section 3.2.

### 3.1    PROBLEM FORMULATION

In this work, we focus on the recognition problem. Let $f_\theta : \mathbb{R}^d \to \mathcal{P}(\mathcal{Y})$ be a classifier that maps a given input $x \in \mathbb{R}^d$ into the probability simplex $\mathcal{P}(\mathcal{Y})$ over the set of labels $\mathcal{Y} = \{1, 2, \ldots, K\}$[1]. In this work, we assume that an input $x$ is a multimodal input. However, in many realistic applications, the input provided to the model might have missing modalities, thus containing either audio only, visual only, or audio-visual information. Let $m \in \{A, V, AV\}$ denote the type of available modality for a given input $x$ corresponding to audio only, visual only, audio-visual, respectively. For a simple formulation, we fix the dimensionality of the input $x$ by replacing the missing modality part with zeros. Further, we assume that $f_\theta$ is a pretrained multimodal model on a training set $\mathcal{D}$. In this work, we make no assumptions on $f_\theta$ (*i.e.* choice of architecture), the dataset $\mathcal{D}$, nor the training process.

At test time, $f_\theta$ is presented with a stream of *unlabeled* data $\mathcal{S}$ with possibly missing modalities. The likelihood with which a certain modality appears in data revealed from $\mathcal{S}$ is characterized by a probability mass function; denoted by $\mathbb{P}_\mathcal{S}(M = m)$. For example, if $\mathbb{P}_\mathcal{S}(M = V) = 0.5$ and $\mathbb{P}_\mathcal{S}(M = AV) = 0.5$, then the audio missing rate in the test stream is $50\%$. In other words, half the data arrives as video only, without its accompanying audio. Let $p_m = \mathbb{P}_\mathcal{S}(M = m)$, then the missing

---

[1]*e.g.* the output after a softmax layer.

rate of different modalities can be equivalently characterized with $P = \{p_A, p_V, p_{AV}\}$. Thus, for a stream with 25% missing video, $P = \{0.25, 0.0, 0.75\}$ (*i.e.* the stream will reveal data with 25% probability of having only audio and 75% probability of revealing both modalities). According to this characterization, one can define the missing rate of at least one modality as $1 - p_{AV}$ with $p_A$ being the rate of missing video and $p_V$ being the audio missing rate. Next, we discuss the online evaluation protocol of $f_\theta$ under the stream $\mathcal{S}$ of unlabeled data.

## 3.2 EVALUATION PROTOCOL

Given our formulation of missing modality in Section 3.1, we are now ready to outline the evaluation protocol. Note that an adaptation method is a function $g(\theta)$ that sequentially adapts the model's parameters $\theta$ to enhance the performance under the missing modality setup. Formally and following the online learning literature (Cai et al., 2021; Ghunaim et al., 2023; Alfarra et al., 2023), we simulate the interaction between the stream $\mathcal{S}$ characterized by $P$ and the TTA method $g$, at each time step $t \in \{0, 1, \ldots, \infty\}$, as follows:

1. $\mathcal{S}$ reveals a sample/batch $x_t$ with its corresponding modality $m$.
2. $f_{\theta_t}$ generates the prediction $\hat{y}_t$.
3. $g$ adapts the model parameter $\theta_t$ to $\theta_{t+1}$.

where $f_{\theta_0}$ is the non-adapted pretrained model. That is, for each revealed sample/batch $x_t$, the model needs to predict its label before receiving the next data point $x_{t+1}$. The adaptation method $g$ can exploit the predicted label to improve the model performance on the next revealed samples. The performance of an adaptation method $g$ is measured in an online manner by comparing the predicted label $\hat{y}_t$ with the ground truth label $y_t$.

## 4 PROPOSED SOLUTION

This section proposes our novel adaptation strategy to combat missing modalities at test time. Recall that the adaptation method $g$ has to satisfy the following requirements. Firstly, $g$ has to be fully self-supervised. The test stream reveals strictly unlabeled data at test time. Secondly, $g$ has to conduct the adaptation in an online manner. That is, $g$ should adapt on each revealed sample/batch of data $x_t$ since $\mathcal{S}$ reveals $x_{t+1}$ only after the model predicts $\hat{y}_t$ (refer to the Evaluation Protocol in Section 3.2).

To formulate our adaptation method, we begin by asking the following question: How should an optimal $f_\theta$ behave under missing modality? We believe that a robust model against missing modality should satisfy two properties. First, the prediction of $f_\theta$ should be invariant to the modality source $m$. Ideally, $f_\theta$ should output the same prediction under both complete and incomplete modality, hence satisfying the following equality: **(i)** $f_\theta(x_i; M = A) = f_\theta(x_i; M = V) = f_\theta(x_i; M = AV) \forall i$. **(ii)** $f_\theta$ should retain high performance in predicting data with complete modality, which is generally satisfied for $f_{\theta_0}$. Satisfying both properties will result in a model that is accurate (satisfying **(ii)**) and robust against missing modality (satisfying **(i)**). To construct an adaptation algorithm that satisfies both properties, we propose to solve the following optimization problem:

$$\theta^* = \arg\min_\theta \mathbb{E}_{x \sim \mathcal{S}} \left[\text{MI}\left(f_\theta(x; m), m\right) + \text{KL}\left(f_\theta(x \mid M = AV) || f_{\theta_0}(x \mid M = AV)\right] \tag{1}$$

where $\text{MI}(u, v)$ is the mutual information between $u$ and $v$ and KL is the KL-Divergence. Note that if the mutual information between two random variables $\text{MI}(u, v) = 0$, then $u$ and $v$ are independent of each other. That is, minimizing the first term in the objective function in equation 1 aims to satisfy property **(i)**. Hence, if $\text{MI}\left(f_{\theta^*}(x; m), m\right) = 0$, then the output of the adapted classifier becomes independent from the available modality at test time. Furthermore, to ensure that the adapted parameters are still performing well upon adaptation, we equip the mutual information minimization with a self-distillation approach through minimizing KL divergence between the prediction of the adapted model and the original $f_{\theta_0}$, satisfying property **(ii)**.

Although the objective function in equation 1 is self-supervised, obtaining $\theta^*$ requires accessing all samples from the stream $\mathcal{S}$ to evaluate the expectation $\mathbb{E}_{x \sim \mathcal{S}}$, which is not available at test time in the online evaluation protocol. To that end, we approximate the expected value during

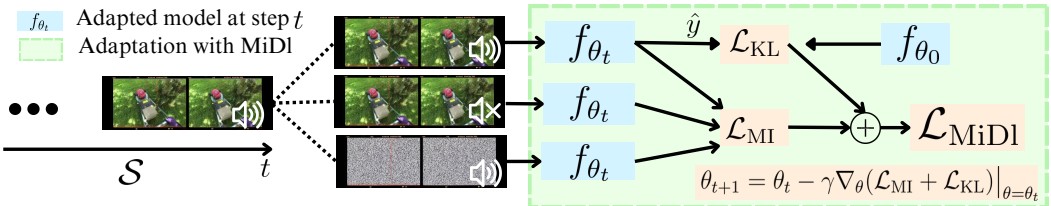

Figure 2: **Adapting at test-time with MiDl.** At test time, the stream reveals a sample. MiDl uses multimodal samples to adapt and requires one forward pass for each modality combination. MiDl leverages (KL) divergence to align the predictions of the adapted model $f_{\theta_t}$ with those of the original model $f_{\theta_0}$, ensuring that the adapted model does not deviate too far from the original model's predictions. The Mutual-Information (MI) component uses the prediction from the different modalities to reduce the dependency on any specific modality, fostering a more generalized and robust prediction across different modality combinations. MiDl updates the model for step $t + 1$ using the combination of KL and MI in Equation 2.

adaptation at time $t$ with the samples $x_t$ revealed from the stream. Hence, our Mutual information with self-Distillation (MiDl) adaptation step at timestamp $t$ can be expressed as the following:

$$\theta_{t+1} = \theta_t - \gamma \nabla_\theta \mathcal{L}_{\text{MiDl}}|_{\theta=\theta_t} = \theta_t - \gamma \nabla_\theta (\mathcal{L}_{\text{MI}} + \mathcal{L}_{\text{KL}})|_{\theta=\theta_t} \quad (2)$$

with $\mathcal{L}_{KL} = \text{KL}\left(f_\theta(x_t; M = AV) || f_{\theta_0}(x_t; M = AV)\right)$

$$\mathcal{L}_{\text{MI}} = \mathbb{E}_m \underbrace{\left[\sum_{i=1}^K f_\theta^i(x_t; m) \log\left(f_\theta^i(x_t; m)\right)\right]}_{\mathcal{L}_{\text{ent}}} - \underbrace{\sum_{i=1}^K \hat{f}_\theta^i(x_t) \log\left(\hat{f}_\theta^i(x_t)\right)}_{\mathcal{L}_{\text{div}}}$$

where $\hat{f}_\theta(x_t) = \mathbb{E}_m[f_\theta(x_t; m)]$, $f_\theta^i(x_t; m)$ is the $i^{th}$ element in the vector $f_\theta(x_t; m)$, and $\gamma > 0$ is the learning rate of the gradient descent step.

To estimate the expectation $\mathbb{E}_m[f_\theta(x_t; m)]$, we conduct three forward passes for $x_t$ with setting $m \in \{A, V, AV\}$. We average these predictions to calculate $\mathcal{L}_{\text{div}}$ and we average their entropy to calculate $\mathcal{L}_{\text{ent}}$ (refer to Figure 2). Note that $\mathcal{L}_{\text{div}}$ is the entropy of the average prediction across modalities. Note that under incomplete modality, $\mathcal{L}_{\text{ent}} = \mathcal{L}_{\text{div}}$ resulting in $\mathcal{L}_{\text{MI}} = 0$. Hence, $\mathcal{L}_{\text{MI}} \neq 0$ only when $x_t$ has complete modalities (refer to Appendix A for details). Based on that, we propose to conduct our adaptation step *only* when $\mathcal{S}$ reveals $x_t$ with complete modalities. However, when $\mathcal{S}$ reveals $x_t$ with incomplete modality, we refrain from adaptation and set $\theta_{t+1} = \theta_t$. This makes the interaction between the stream $\mathcal{S}$ and our proposed MiDl at each time step $t$ take the following form:

> 1. $\mathcal{S}$ reveals a sample/batch $x_t$ with its corresponding modality $m$.
> 2. $f_{\theta_t}$ generates the prediction $\hat{y}_t$.
> 3. If $x_t$ is with complete modalities then $g$ adapts the model parameter $\theta_t$ to $\theta_{t+1}$ through equation 2, else set $\theta_{t+1} = \theta_t$.

For $x_t$ with missing modality, we leverage the most recent adapted model to perform predictions without adaptation. Since this work focuses on the multimodal setting, we assume that $p_{AV} \neq 0$. This setup aligns with real-world scenarios where multimodal data streams are common, and some modal-complete instances are expected. Note that adapting a multimodal model at test time to an unimodal stream is particularly challenging without labeled data. Nevertheless, to demonstrate that MiDl does not degrade the original multimodal model's performance in this extreme case, we also report results with $p_{AV} = 0$.

**Takeaway.** We are the first to formulate the missing modality challenge as a test-time adaptation problem. Our work makes no assumptions about the training phase in terms of architecture or training objectives. We only require a model that works with multiple modalities at test time. We proposed MiDl, the first test-time adaptation method that effectively combats the missing modality challenge. MiDl updates the model only when it encounters modality-complete samples but generates predictions for all samples, regardless of the modalities they contain. MiDl operates on **any pretrained**

**multimodal** model at **test-time** by adapting its parameters on the received **unlabeled** data in an **unsupervised** online manner.

## 5 EXPERIMENTS

We thoroughly analyze MiDl's performance under different missing modality scenarios. We present our experimental setup in Section 5.1. In Section 5.2, we compare MiDl on a simple TTA scenario where the model has to adapt while replying to the stream. In Section 5.3, we study MiDl's behavior under the Long-Term Adaptation (LTA) setup that happens when the model has been exposed to a long stream of data $\mathcal{S}$. Finally, Section 5.4 analyzes the scenario, in which one has access to some unlabeled data from an out-of-domain source before the model is deployed. Under all these three scenarios, we show that MiDl is the better alternative to combating missing modalities at test time.

### 5.1 SETUP

We follow Ramazanova et al. (2024) and use validation sets of Epic-Kitchens and Epic-Sounds and report action accuracy. We use Ego4D-AR videos for the out-of-domain experiments in Section 5.4.

**Datasets.** **Epic-Kitchens** (Damen et al., 2018; 2022) contains 100 video hours of people performing cooking activities recorded with wearable cameras. The dataset is commonly used for benchmarking audiovisual egocentric action recognition. Each instance is annotated as a noun and verb pair (*e.g.,* cut tomato), with a total of 300 noun and 97 verb classes. The validation set contains 9668 samples. **Epic-Sounds** (Huh et al., 2023) provides sound-based annotations for the same 100 video hours. It has 44 classes and 8045 validation samples. We stick to the official train/val/test splits provided for both datasets. The approximate ratios are 75% for training, 10% for validation, and 15% for testing.

We assess the effectiveness of the baselines and our proposed MiDl in combating missing modalities at test time. To do so, we present the pretrained model with a stream $\mathcal{S}_{\text{val}}$ of unlabeled validation data where we drop one modality with a rate of $1 - p_{AV}$. We set $p_{AV} \in \{0.0, 0.25, 0.5, 0.75, 1.0\}$ resulting in a missing rate of $\{100\%, 75\%, 50\%, 25\%, 0\%\}$, respectively. Following Ramazanova et al. (2024), for each dataset, we drop the primary modality (*i.e.* sound for Epic-Sounds and video for Epic-Kitchens). Thus, for a 75% missing rate (*i.e.* $p_{AV} = 0.25$), the stream containing Epic-Sounds has $P = \{0.0, 0.75, 0.25\}$, and the one for Epic-Kithcnes has $P = \{0.75, 0.0, 0.25\}$. We also report the performance of unimodal models that rely solely on the available modality (e.g., video for Epic-Sounds and audio for Epic-Kitchens). Ideally, a well-adapted multimodal model should match or exceed this performance.

**Architecture.** Unless stated otherwise, we use the stronger architecture Multimodal Bottleneck Transformer (MBT) (Nagrani et al., 2021) as $f_\theta$. Yet, we experiment with the vanilla self-attention architecture (Vaswani et al., 2017) in Section 6.1. Each backbone is fine-tuned on the training set of the corresponding dataset.

### 5.2 MiDl IMPROVES PERFORMANCE

Table 1 compares our proposed MiDl against three off-the-shelf TTA methods. Namely, we include the information maximization approach, Shot (Liang et al., 2020), and the state-of-the-art entropy minimization with data selection from ETA (Niu et al., 2022) (refer to Section B.1 for details).

We observe that: **(i)** MiDl significantly enhances the baseline performance under missing modality. We record a significant gain of 5% and 7% on Epic-Kitchens with missing rates of 50% and 75%, respectively. In Epic-Sounds, MiDl boosts the accuracy of the baseline from 37.1% and 28.3% to 38.8% and 29.8% under the same missing rates, respectively. Note that this performance boost comes at no cost of retraining the model; simply adapting it during test time. This result demonstrates how our proposed mutual information minimization encourages the model to become invariant to domain shifts due to missing modality. **(ii)** MiDl successfully retains the baseline performance when all modalities are present, with accuracies of 55.0% and 63.7% at 0% missing rate on Epic-Sounds and Epic-Kitchens, respectively. This demonstrates that our proposed KL divergence regularization effectively preserves the information retention capability of the baseline under modal-complete

Table 1: **Combating missing modalities at test time.** The first two rows show the unimodal performance and the MBT baseline with no adaptation. We show three alternative TTA methods and demonstrate that our proposed MiDl is effective at combating missing modalities at test time, outperforming all presented TTA baselines. Refer to Table 11 to see the standard deviations.

| $1-p_{AV}$ Model | Epic-Sounds (%) | | | | | Epic-Kitchens (%) | | | | |
|---|---|---|---|---|---|---|---|---|---|---|
| | 0 | 25 | 50 | 75 | 100 | 0 | 25 | 50 | 75 | 100 |
| Unimodal | 41.4 | 41.4 | 41.4 | 41.4 | 41.4 | 40.0 | 40.0 | 40.0 | 40.0 | 40.0 |
| Baseline | 55.1 | 45.6 | 37.1 | 28.3 | 19.5 | 63.9 | 55.5 | 46.8 | 37.9 | 29.5 |
| +Shot | 55.0 | 45.6 | 37.1 | 28.5 | **20.0** | **63.9** | 55.9 | 47.9 | 40.6 | **34.3** |
| +Tent | 54.8 | 45.0 | 35.9 | 26.5 | 17.8 | 63.7 | 54.0 | 39.2 | 24.2 | 9.9 |
| +ETA | **55.1** | 45.6 | 37.1 | 28.3 | 19.5 | 63.5 | 51.3 | 33.7 | 20.6 | 7.9 |
| +MiDl (ours) | 55.0 | **46.8** | **38.8** | **29.8** | 19.5 | 63.7 | **58.4** | **52.4** | **46.4** | 29.5 |

Table 2: **Adaptation at Test-time under Long-term Adaptation and with Ego4D warm-up. LTA.** We showcase the results of MiDL under the assumption that the stream of data is very long. We use unlabeled data to simulate a longer stream and report results on the validation set of each dataset. Our MiDl benefits from long-term adaptation, especially at higher missing rates (>75%). **Ego4D warm-up.** We show another use cause of MiDL, in which the assumption is having access to out-of-domain unlabeled data to adapt before deployment. The results showcase MiDL's capabilities on leveraging unlabeled-out-of-domain data to combat missing modalities.

| $1-p_{AV}$ Model | Epic Sounds (%) | | | | | Epic Kitchens (%) | | | | |
|---|---|---|---|---|---|---|---|---|---|---|
| | 0 | 25 | 50 | 75 | 100 | 0 | 25 | 50 | 75 | 100 |
| Baseline | **55.1** | 45.6 | 37.1 | 28.3 | 19.5 | **63.9** | 55.5 | 46.8 | 37.9 | 29.5 |
| +MiDl | 55.0 | 46.8 | 38.8 | 29.8 | 19.5 | 63.7 | **58.4** | **52.4** | 46.4 | 29.5 |
| + MiDl - LTA | 54.9 | **46.8** | **39.5** | **32.6** | **26.0** | 63.7 | **58.4** | **52.4** | **46.7** | **41.4** |
| + Ego4D Warm-up | 55.0 | 46.5 | 38.6 | 30.4 | 20.4 | 63.7 | **58.4** | **52.4** | **46.7** | 37.8 |

inference. **(iii)** We also observe that presented TTA methods are less effective. This limitation arises because TTA methods are designed to tackle covariate domain shifts and, thus, are not tailored to enhance performance under this specific type of domain shift (missing modality).

## 5.3 PERFORMANCE UNDER LONG-TERM ADAPTATION (LTA)

Next, we analyze the effectiveness of our proposed MiDl under a long stream of data $\mathcal{S}$. Since MiDl operates at test time, its performance gain can vary depending on the amount of data revealed at test time. Note that for any $p_{AV} \neq 0$, as $t \to \infty$, MiDl would be exposed to a large amount of unlabeled data with complete modality; enhancing the invariance properties of the adapted model against missing modality.

To study this interesting setting, we present MiDl with $\mathcal{S}_{in}$ followed by $S_{val}$ used in Section 5.2. In this scenario, we allow MiDl to access unlabeled data with complete modality from $\mathcal{S}_{in}$, to *only* perform adaptation. Then, we assess the efficacy of MiDl by performing adaptation and evaluation on $S_{val}$. We then ask the following question: how would MiDl perform after this long adaptation? We let $\mathcal{S}_{in}$ be a subset of training data, and test the model on $\mathcal{S}_{val}$ being the validation set, following the standard evaluation in Section 5.2.

Table 2 summarizes the results on Epic-Sounds and Epic-Kitchens datasets at the same missing rates considered in Section 5.2. We observe: **(iv)** the longer the stream is, MiDl provides a bigger performance gain. For example, MiDl further improves the non-adapted baseline by 4.3% and 8.8% on Epic-Sounds and Epic-Kitchens, respectively, under a missing rate of 75% (*i.e.* $p_{AV} = 0.25$). In addition, even under 100% missing rate, MiDl improves the accuracy by a notable margin of 6.5% and 11.9% on Epic-Sounds and Epic-Kitchens, respectively. That is, the adaptation on $\mathcal{S}_{in}$ unlocks a bigger potential of MiDl for life-long adaptation even when $\mathcal{S}_{val}$ reveals data with a single modality. **(v)** Unlike MiDl, other adaptation approaches do not benefit from this long stream setup as their objective functions do not promote building an invariant model against missing modality. We present these results in Table 7.

## 5.4 WARM-UP ON EGO4D: EXPLOITING OUT OF DOMAIN ADAPTATION

Next, we analyze another practical setup in which the model can be warmed up with some available data before deployment. In particular, we consider the case when not only the pretrained model $f_\theta$ is provided, but unlabeled data from out-of-domain, denoted as $\mathcal{S}_{out}$ can also be accessible. Given this setup, we wonder: would warming up with MiDl on a different data distribution $\mathcal{S}_{out}$ help combat missing modalities in $\mathcal{S}_{val}$?

To answer this question, we leverage the recent Ego4D (Grauman et al., 2022) data. Although Ego4D has egocentric videos, they come from very different environments and scenarios that deviate from the usual kitchen scene. These changes introduce additional domain shifts when evaluating on Epic-Sounds and Epic-Kitchens. We set $\mathcal{S}_{out}$ to be 5,000 clips of the Ego4D-AR training set. It is worth noting that we keep using our self-supervised MiDL objective, and we do not require any labels from Ego4D-AR. We use our setup from Section 5.3 and perform adaptation on $\mathcal{S}_{out}$ followed by the standard evaluation (Section 5.2) on $\mathcal{S}_{val}$. We refer to the adaptation on $\mathcal{S}_{out}$ as the warm-up phase.

Table 2 summarizes the results where we show the performance of the non-adapted baseline, our proposed MiDl when adapted solely on $\mathcal{S}_{val}$, and our MiDL equipped with warm-up adaptation on $\mathcal{S}_{out}$. We observe that **(vi)** conducting a warm-up generally positively influences overall performance in cases of missing modality. The enhanced version of MiDl with Ego4D warm-up improves over MiDl by 0.6% and 0.3% on Epic-Sounds and Epic-Kitchens, respectively, under a missing rate of 75%. Furthermore, we observe that even under 100% missing rate, adapting on $\mathcal{S}_{out}$ enhances the accuracy of MiDl by an impressive 8% on Epic-Kitchens. This demonstrates the versatility of MiDl, which provides consistent performance gains under different setups.

**Takeaway.** In this section, we showcased the effectiveness of our proposed MiDl in combating the missing modality challenge at test time (Section 5.2). Further, we analyzed the impact of long-term adaptation where MiDl provided further performance gain where the relative improvement is up to 30% (Section 5.3). At last, we showed how adapting with MiDl on out-of-distribution data, mimicking data scarcity situations, still boosts the model's accuracy (Section 5.4).

## 6 ANALYSIS ON MIDL

In this section, we conduct a comprehensive analysis of MiDl. In particular, we show how our proposed test-time adaptation is agnostic to the choice of $f_\theta$ (Section 6.1), the missing modality (Section 6.2). We conclude by analyzing the different components of MiDl in Section 6.4 and its computational requirements in Section 6.5.

### 6.1 AGNOSTIC TO ARCHITECTURE CHOICE

Here, we analyze the robustness of MiDl against the architecture choice $f_\theta$. To this end, we replicate our experimental setup from Sections 5.2 and 5.3 but set $f_\theta$ to be the multimodal fusion via vanilla self-attention architecture (Vaswani et al., 2017), as opposed to the MBT architecture (Nagrani et al., 2021).

Table 3 summarizes the results under different missing rates. We compare the non-adapted baseline, Shot and ETA, and our proposed MiDl. We observe that MiDl provides a consistent performance boost under the self-attention architecture, similar to our observations with the MBT architecture in Section 5.2. For example, MiDl improves the accuracy of the baseline under 50% missing rate by a notable 1.1% without affecting the performance under complete modality.

Table 3: **MiDL performance with self-attention baseline.** We showcase the effectiveness of MiDL with multimodal self-attention. MiDL enhances performance across all missing rates, underscoring its robustness and adaptability to various underlying architectures.

| Model $\quad 1 - p_{AV}$ | Epic-Sounds (%) | | | | |
|---|---|---|---|---|---|
| | 0 | 25 | 50 | 75 | 100 |
| Self-Att. Baseline | 45.3 | 38.8 | 32.7 | 26.7 | 20.5 |
| +Shot | 45.5 | 39.0 | 32.8 | 26.8 | 20.7 |
| +Tent | 45.3 | 38.6 | 32.3 | 26.0 | 19.8 |
| +ETA | 45.3 | 38.8 | 32.7 | 26.7 | 20.5 |
| +MiDl (ours) | **45.5** | 39.5 | 33.8 | 27.5 | 20.5 |
| +MiDl - LTA (ours) | **45.5** | **39.6** | **34.5** | **29.0** | **23.2** |

Table 4: **Adaptation at Test-time - Other Missing Modalities**. In this table we show the results using the complementary modality for each of the datset, *i.e.* video for Epic Sounds and Audio for Epic Kitchens. We observe that MiDL improves consistently under this setup, highlighting its robustness to different types of modalities missing at test time.

| $1 - p_{AV}$ Model | Epic Sounds (%) | | | | | Epic Kitchens (%) | | | | |
|---|---|---|---|---|---|---|---|---|---|---|
| | 0 | 25 | 50 | 75 | 100 | 0 | 25 | 50 | 75 | 100 |
| Unimodal | 46.5 | 46.5 | 46.5 | 46.5 | 46.5 | 63.2 | 63.2 | 63.2 | 63.2 | 63.2 |
| Baseline | **55.1** | 53.4 | 51.8 | 50.5 | 48.8 | **63.9** | 61.0 | 58.0 | 54.8 | 52.1 |
| +MiDl | 55.0 | 53.3 | 51.8 | 50.7 | 48.8 | 63.7 | 61.6 | 59.1 | 55.9 | 52.1 |
| +MiDl - LTA | 55.0 | **53.4** | **52.0** | **51.0** | **49.4** | 63.7 | **61.7** | **59.5** | **57.3** | **55.3** |

Addtionally, Table 3 presents the results under Long-Term Adaptation (LTA), following the setup in Section 5.3. Similar to our earlier findings, LTA unlocks a better potential of MiDl where further performance gain is attained. Under this setup, MiDl improves the baseline by 2% under 50% and 75% missing rates, and by 2.7% under 100% missing rates. These results show that MiDl is agnostic to the choice of architecture.

## 6.2 AGNOSTIC TO THE TYPE OF MISSING MODALITY

In previous sections, we focused our experiments on the effectiveness of MiDl when the dominant modality is missing. In particular, we analyzed dropping the audio modality for Epic-Sounds and the visual modality for Epic-Kitchens. In this section, we attempt to study the robustness of MiDl against dropping the non-dominant modality at test time. Thus, in these experiments, we drop the visual modality in Epic-Sounds and audio in Epic-Kitchens. In contrast to the scenarios with missing dominant modality, these baselines experience less performance degradation. We replicate our experimental setup from Section 5.3 where we compare the non-adapted baseline, its equipped version with our proposed MiDl, and the LTA effect of MiDl.

Table 4 summarizes the results. Our experiments show that MiDl consistently enhances the performance of the pretrained model irrespective of the type of missing modality. For example, MiDl improves the baseline by 1% on Epic Kitchens under missing rates of 50% and 75%. Further, and similar to our observations in Sections 5.3 and 6.1, the LTA further improves MiDl by over 3% under 100% missing rate (*i.e.* $p_{AV} = 0$). In Table 9, we demonstrate how MiDl generalizes to a "mixed modality" setup, where either modality could be missing at test time.

## 6.3 AGNOSTIC TO PRETRAINING

In previous sections, we adopted the baseline from Ramazanova et al. (2024), which uses masked autoencoder pretraining. To demonstrate how MiDl can generalize to the different pretraining mechanisms, we apply MiDl on the Omnivore backbone (Girdhar et al., 2022). Note that Omnivore is a modality-agnostic vision model with leverages several visual modalities in one unified architecture during pretraining. During testing, Omnivore is used for single-modality downstream tasks on any visual modality (Girdhar et al., 2022). Thus, we use omnivore to initialize each one of our backbones. We report the results in Table 5. We observe how MiDl provides a significant performance boost over the Omnivore baseline, when presented with missing modalities at test time. For example, under the 25% missing rate, MiDl improves the baseline performance by 9.5%, from 48.1% to 57.6%. These results demonstrate that MiDl maintains its robustness and adaptability across different pretraining strategies, significantly improving the Omnivore baseline. This, yet again, highlights MiDl's generalization capability.

## 6.4 COMPONENTS OF MIDL

At last, we ablate the effectiveness of each component of MiDl. Recall from our adaptation step in equation 1 that MiDl has two components: mutual information minimization (Mi) and self-distillation through minimizing KL-divergence (Dl). We believe that the success of MiDl is attributed to the interplay between both components.

Table 5: **MiDl performance with Omnivore pretraining.** MiDl is highly effective when applied to Omnivore model, demonstrating its effectiveness with a different pretraining strategy.

| $1 - p_{AV}$ Model | Epic-Kitchens (%) | | | | |
|---|---|---|---|---|---|
| | 0 | 25 | 50 | 75 | 100 |
| Omnivore Baseline | **65.6** | 48.1 | 47.6 | 46.0 | **44.2** |
| +MiDl (ours) | **65.6** | **57.6** | **52.4** | **47.5** | **44.2** |

Table 6: **Analyzing MiDl components.** We analyze the different components of MiDL. When the Mutual-Information (MI) component is missing, the model does not have any reason to adapt since the KL divergence is maximized by predicting the same as the base model. When KL is not present, the MI alone deviates from the initial results and performs poorly under higher missing rates.

| $1 - p_{AV}$ Model | $\mathcal{L}_{\text{MI}}$ | $\mathcal{L}_{\text{KL}}$ | Epic Sounds (%) | | | | Epic Kitchens (%) | | | |
|---|---|---|---|---|---|---|---|---|---|---|
| | | | 0 | 25 | 50 | 75 | 0 | 25 | 50 | 75 |
| Baseline | ✗ | ✗ | 55.1 | 45.6 | 37.1 | 28.3 | **63.9** | 55.5 | 46.8 | 37.9 |
| + Dl | ✗ | ✓ | **55.2** | 45.6 | 37.1 | 28.3 | 63.9 | 55.5 | 46.8 | 37.9 |
| + Mi | ✓ | ✗ | 40.4 | 39.3 | 36.1 | 29.6 | 53.5 | 50.5 | 47.6 | 45.9 |
| +MiDl (ours) | ✓ | ✓ | 55.0 | **46.8** | **38.8** | **29.8** | 63.7 | **58.4** | **52.4** | **46.4** |

To analyze the importance of each component, we adapt the baseline with each loss function independently and compare the performance. Table 6 summarizes the results. We observe that adapting solely with self-distillation (*i.e.* minimizing $\mathcal{L}_{\text{KL}}$) results in no adaptation. On the contrary, adaptation by minimizing only $\mathcal{L}_{\text{MI}}$ can result in a significant performance drop for low missing rates. While minimizing $\mathcal{L}_{\text{MI}}$ can indeed result in $f_\theta$ that is robust against missing modality; it might be less accurate for modal-complete samples (*i.e.* satisfying the first **(i)** property in Section 4 and violating the second **(ii)**). Note that for high missing rates, $\mathcal{L}_{\text{MI}}$ can result in a performance boost as a small number of adaptation steps will enhance the invariance properties while not causing significant divergence from the original model. MiDl balances information minimization with the information retention loss $\mathcal{L}_{\text{KL}}$, providing consistent performance gains under all missing rates.

## 6.5 COMPUTATIONAL REQUIREMENT OF MIDL

While our results have demonstrated the efficacy of MiDl in providing performance gains, we note that this improvement comes at a computational cost. In fact, conducting an adaptation step with MiDl requires 3 inferences through $f_{\theta_t}$ and an inference through the initial pretrained model $f_{\theta_0}$, followed by a single backward pass. This makes an inference through MiDl $5\times$ more expensive than doing inference without any adaptation. In practice, the latency of MiDl is only $2\times$ slower than the non-adapted model since all the additional 4 forward passes can be performed in parallel.

**Takeaway.** In this section, we conducted comprehensive analysis on our proposed MiDl. We showed that MiDl is agnostic to the choice of architecture (Section 6.1) and the type of missing modality (Section 6.2), and the type of pre-training (Section 6.3). We further analyzed the importance of both components of MiDl in Section 6.4 and its computational requirements in Section 6.5. MiDl shows remarkable performance across the board, consistently delivering strong results regardless of the dataset or scenario.

## 7 CONCLUSIONS

In this work, we presented MiDl, a new method for improving how pretrained video recognition models handle missing modalities at test time. MiDl improves the model's ability to give accurate predictions regardless of the availability of modalities by minimizing mutual information and using self-distillation. Our experiments show that MiDl can significantly increase accuracy across various datasets and scenarios and under various missing rates, making it a practical solution for real-world applications dealing with incomplete modalities.

ACKNOWLEDGEMENTS

The research reported in this publication was supported by funding from KAUST Center of Excellence on GenAI, under award number 5940.

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

## A    PROPOSED SOLUTION: EXTENDED DISCUSSION

In Section 4, we proposed our novel adaptation strategy; MiDl. First, note that our intuition of minimizing the mutual information between the output of the network and the modality source comes from the following observation: Let $X$ and $Y$ be two random variables, then

$$\text{If } \text{MI}(X, Y) = 0 \; \rightarrow \; X \text{ and } Y \text{ are independent.}$$

$$\textit{Proof: } \text{MI}(X, Y) = \sum_{x \in X} \sum_{y \in Y} P_{XY}(x, y) \log \left( \frac{P_{XY}(x, y)}{P_X(x) P_Y(y)} \right)$$

$$\text{If } \text{MI}(X, Y) = 0 \; \rightarrow \; P_{XY}(x, y) = P_X(x) P_Y(y)$$

That is, minimizing the mutual information between the output prediction of $f_\theta$ and the available modality should make the adapted network robust against missing modality. Second, MiDl adapts the pretrained model $f_\theta$ on the unlabeled data revealed from the stream only if the revealed data is with complete modality. This is since: **(1)** the KL-divergence loss only applies on data with full modality and **(2)** the information minimization loss only operates under complete modality. To wit, let $x_t$ be revealed with $m = A$. Then the estimate of our $\mathcal{L}_{\text{MI}}$ will be

$$\mathcal{L}_{\text{MI}} = \underbrace{\sum_{i=1}^{K} f_\theta^i(x_t; M = A) \log \left( f_\theta^i(x_t; M = A) \right)}_{\mathcal{L}_{\text{ent}}}$$

$$- \underbrace{\sum_{i=1}^{K} f_\theta^i(x_t; M = A) \log \left( f_\theta^i(x_t; M = A) \right)}_{\mathcal{L}_{\text{div}}} = 0.$$

This is since $\mathbb{E}_m$ will be estimated with a single point at $m = A$. Thus, MiDl adapts the parameters only when $\mathcal{S}$ reveals data with complete modality at test time.

## B    EXPERIMENTS

### B.1    IMPLEMENTATION DETAILS

In Section 5.1, we detailed our experimental setup. Here, and for reproducibility, we provide the implementation details for MiDl and the baselines. Note that for all considered adaptation methods, we follow the standard practice in the test-time adaptation literature (Wang et al., 2020; Liang et al., 2020; Niu et al., 2022; Niu14 et al., 2023) and only adapt the learnable parameters of the normalization layers. We freeze the rest of the model parameters during the update step. Further, for MiDl, we balance the mutual information loss and the self-distillation loss through:

$$\mathcal{L}_{\text{MiDl}} = \lambda_1 \mathcal{L}_{\text{MI}} + \lambda_2 \mathcal{L}_{\text{KL}}.$$

Note that we set $\lambda_1 = \lambda_2 = 3$ for all our experiments. These hyperparameters were determined through a grid search to identify the optimal settings for the task. Further, we conduct the adaptation step with an SGD (Robbins & Monro, 1951) step with a learning rate of $25 \times 10^{-4}$, and a momentum of $0.9$, following (Niu14 et al., 2023; Niu et al., 2022; Wang et al., 2020).

Regarding the considered test-time adaptation baselines, we considered the entropy minimization approach known as Tent (Wang et al., 2020), its improved version equipped with data-selection process ETA (Niu et al., 2022), and the information maximization Shot (Liang et al., 2020). We followed the official implementation of each method and used the recommended hyperparameters.

Each experiment was run using one V100 GPU. We repeat all experiments 5 times with different seeds and report the average accuracy.

### B.2    BASELINES UNDER LONG TERM ADAPTATION

We extend our comparison against the considered baselines under the Long Term Adaptation (LTA) setup. We replicate our experimental setup in Section 5.3 and compare MiDl against SHOT, Tent, and

Table 7: **Adaptation at Test-time under Long-term Adaptation.** We showcase the results of MiDL under the assumption that the data stream is very long. We use unlabeled data to simulate a longer stream and report results on the validation set of each dataset. Our MiDl benefits from long-term adaptation. Especially at higher missing rates (>75%).

| $1 - p_{AV}$ Model | Epic-Sounds (%) | | | | | Epic-Kitchens (%) | | | | |
|---|---|---|---|---|---|---|---|---|---|---|
| | 0 | 25 | 50 | 75 | 100 | 0 | 25 | 50 | 75 | 100 |
| Unimodal | 41.4 | 41.4 | 41.4 | 41.4 | 41.4 | 40.0 | 40.0 | 40.0 | 40.0 | 40.0 |
| Baseline | **55.1** | 45.6 | 37.1 | 28.3 | 19.5 | **63.9** | 55.5 | 46.8 | 37.9 | 29.5 |
| +Shot - LTA | 55.0 | 45.6 | 37.2 | 28.7 | 20.3 | 63.8 | 56.0 | 48.2 | 41.0 | 34.4 |
| +Tent - LTA | 54.5 | 44.6 | 35.5 | 26.1 | 17.7 | 62.7 | 53.5 | 40.0 | 25.8 | 12.3 |
| +ETA - LTA | 55.0 | 45.5 | 37.0 | 28.2 | 19.5 | 60.9 | 48.8 | 33.0 | 19.3 | 7.5 |
| +MiDl - LTA (ours) | 54.9 | **46.8** | **39.5** | **32.6** | **26.0** | 63.7 | **58.4** | **52.4** | **46.7** | **41.4** |

Table 8: **Adaptation at Test-time - Updating all parameters**. We show the results when we unfreeze all network parameters, not only the normalization layer. We observe that there is no significant difference when compared to updating only the normalization layers.

| $1 - p_{AV}$ Model | Epic-Kitchens (%) | | | | |
|---|---|---|---|---|---|
| | 0 | 25 | 50 | 75 | 100 |
| Unimodal | 40.0 | 40.0 | 40.0 | 40.0 | 40.0 |
| Baseline | 63.9 | 55.5 | 46.8 | 37.9 | 29.5 |
| +MiDl (all parameters) | 63.6 | **58.4** | 52.4 | 46.3 | 29.5 |
| +MiDl (norm layers) | **63.8** | **58.4** | 52.1 | 44.9 | 29.5 |
| +MiDl - LTA (all parameters) | 63.6 | 58.3 | **52.5** | **47.1** | **42.0** |
| +MiDl - LTA (norm layers) | **63.8** | **58.4** | 52.4 | 46.7 | 41.4 |

ETA. We report the results in Table 7 for the MBT architecture. We observe consistent findings in Section 5.3, where naive adaptation baselines do not benefit from this long-term adaptation. Further, we find that MiDl is the only adaptation method that provides further performance gains under this LTA setup.

Table 9: **MiDL Performance with Mixed Modalities Setup.** We present results similar to those in Table 4 for Epic Sounds, but this time under mixed missing modalities at test time. Our results demonstrate that MiDL continues to enhance the base model's performance even in the challenging scenario where any modality may be absent.

| $1 - p_{AV}$ Model | Epic-Sounds (%) | | | | |
|---|---|---|---|---|---|
| | 0 | 25 | 50 | 75 | 100 |
| Baseline | 55.1 | 49.5 | 44.0 | 39.5 | 34.1 |
| +MiDl (ours) | 55.1 | 50.0 | 45.0 | 40.3 | 34.1 |
| +MiDl - LTA (ours) | 55.0 | **50.3** | **45.4** | **42.1** | **37.4** |

### B.3 ADAPTING MORE LAYERS

At last, we extend our analysis on MiDl to include the effect of adapting more parameters. In particular, we compare adapting only the learnable parameters of the normalization layers against adapting the whole network parameters. We report the result in Table 8 on Epic-Kitchens. We observe that adapting all network parameters with MiDl results in a minor performance gain. For instance, under the 75% missing rate, adapting all parameters improves over adapting only the normalization layers by 0.4% under the LTA setup and by 1.4% with the test-time adaptation of MiDl. We note

Table 10: **KL loss on each prediction.** We apply KL loss to the prediction of each modality. As the audiovisual predictions are derived from the individual modality predictions, there is no much difference in the performance.

| $1 - p_{AV}$ Model | Epic-Sounds (%) | | | | | Epic-Kitchens (%) | | | | |
|---|---|---|---|---|---|---|---|---|---|---|
| | 0 | 25 | 50 | 75 | 100 | 0 | 25 | 50 | 75 | 100 |
| Unimodal | 41.4 | 41.4 | 41.4 | 41.4 | 41.4 | 40.0 | 40.0 | 40.0 | 40.0 | 40.0 |
| Baseline | **55.1** | 45.6 | 37.1 | 28.3 | 19.5 | **63.9** | 55.5 | 46.8 | 37.9 | 29.5 |
| +MiDl (ours) | 55.0 | **46.8** | 38.8 | 29.8 | 19.5 | 63.7 | **58.4** | **52.4** | **46.4** | 29.5 |
| +MiDl (KL on each modality) | 55.0 | **46.8** | **38.9** | **30.0** | 19.5 | 63.5 | 57.9 | 52.1 | **46.4** | 29.5 |

Table 11: **Combating missing modalities at Test-time.** We present the extended results of Table 1 that include the standard deviation of each of the results. Our proposed MiDl is effective at combating missing modalities at test time, outperforming all presented TTA baselines by convincing margins over several runs.

| $1 - p_{AV}$ Model | Epic-Sounds (%) | | | | | Epic-Kitchens (%) | | | | |
|---|---|---|---|---|---|---|---|---|---|---|
| | 0 | 25 | 50 | 75 | 100 | 0 | 25 | 50 | 75 | 100 |
| Unimodal | 41.4 | 41.4 | 41.4 | 41.4 | 41.4 | 40.0 | 40.0 | 40.0 | 40.0 | 40.0 |
| Baseline | 55.1 | 45.6 | 37.1 | 28.3 | 19.5 | 63.9 | 55.5 | 46.8 | 37.9 | 29.5 |
| +Shot | 55.0±0.04 | 45.6±0.02 | 37.1±0.06 | 28.5±0.06 | **20.0**±0.07 | **63.9**±0.04 | 55.9±0.05 | 47.9±0.1 | 40.6±0.08 | **34.3**±0.1 |
| +Tent | 54.8±0.04 | 45.0±0.08 | 35.9±0.06 | 26.5±0.04 | 17.8±0.05 | **63.7**±0.07 | 54.0±0.15 | 39.2±0.25 | 24.2±0.24 | 9.9±0.22 |
| +ETA | **55.1**±0.02 | 45.6±0.02 | 37.1±0.01 | 28.3±0.00 | 19.5±0.00 | 63.5±0.04 | 51.3±0.75 | 33.7±0.26 | 20.6±0.24 | 7.9±0.27 |
| +MiDl (ours) | 55.0±0.10 | **46.8**±0.09 | **38.8**±0.11 | **29.8**±0.07 | 19.5 | **63.7**±0.08 | **58.4**±0.04 | **52.4**±0.05 | **46.4**±0.11 | **29.5** |

here that this comes at a computational expense as it is faster and more efficient to adapt only the normalization layers.

### B.4 MIXED MODALITIES

In this setup we also set $p_{AV} \in \{0.0, 0.25, 0.5, 0.75, 1.0\}$, but also $p_A = p_V = 0.5 * (1 - p_{AV})$. Thus, for each missing rate ($1 - p_{AV}$), exactly half of the modal-incomplete samples have missing audio, and the other half have missing video. The results of this setup are shown in Table 9. We observe that MiDl consistently improves the baseline performance for all missing rates. Furthermore, when presented with a long stream (LTA), MiDl benefits by further improving the accuracy.

### B.5 KL LOSS ON ALL PREDICTIONS

One might suggest applying KL loss to both the individual audio and video predictions, as well as the combined audiovisual predictions. We present the results of this experiment in Table 10. We found that applying KL loss individually to each modality produced results very similar to applying it to the combined audiovisual predictions. This is because the combined predictions essentially average the individual modality predictions, making the effect of applying the loss individually or in combination nearly equivalent.

## C LIMITATIONS

As we mentioned in Section 6.5, MiDl requires three forward passes of the model over a single instance, which can be implemented efficiently by parallelizing the pass on the GPU. However, it does require more FLOPs than a method without adaptation. Additionally, MiDl can only adapt to modal-complete instances. Finally, our experiments are limited to audiovisual egocentric datasets. Although we designed MiDl without these constrains in mind, the validation of our method was only made under this setup. Thus, results on other multimodal datasets with other modalities and other data sources need further validation. However, we believe that MiDl would still work in other scenarios.

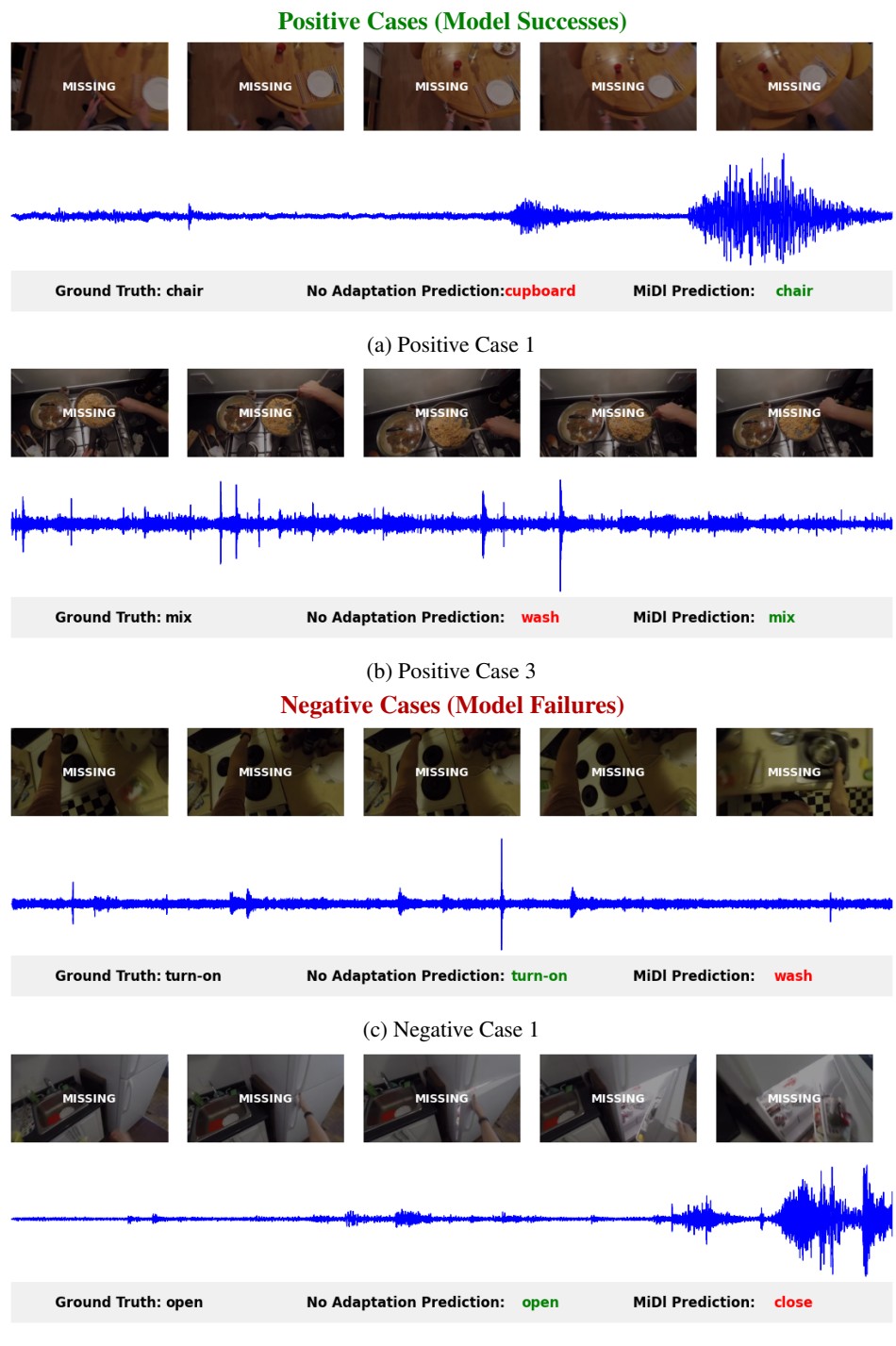

Figure 3: **Qualitative analysis of MiDl's adaptation performance on Epic-Kitchens.** The top two subfigures highlight **positive cases** where MiDl successfully adapts to predict the correct label (marked in green). Conversely, the bottom two subfigures illustrate **negative cases** (marked in red) where adaptation introduces errors.

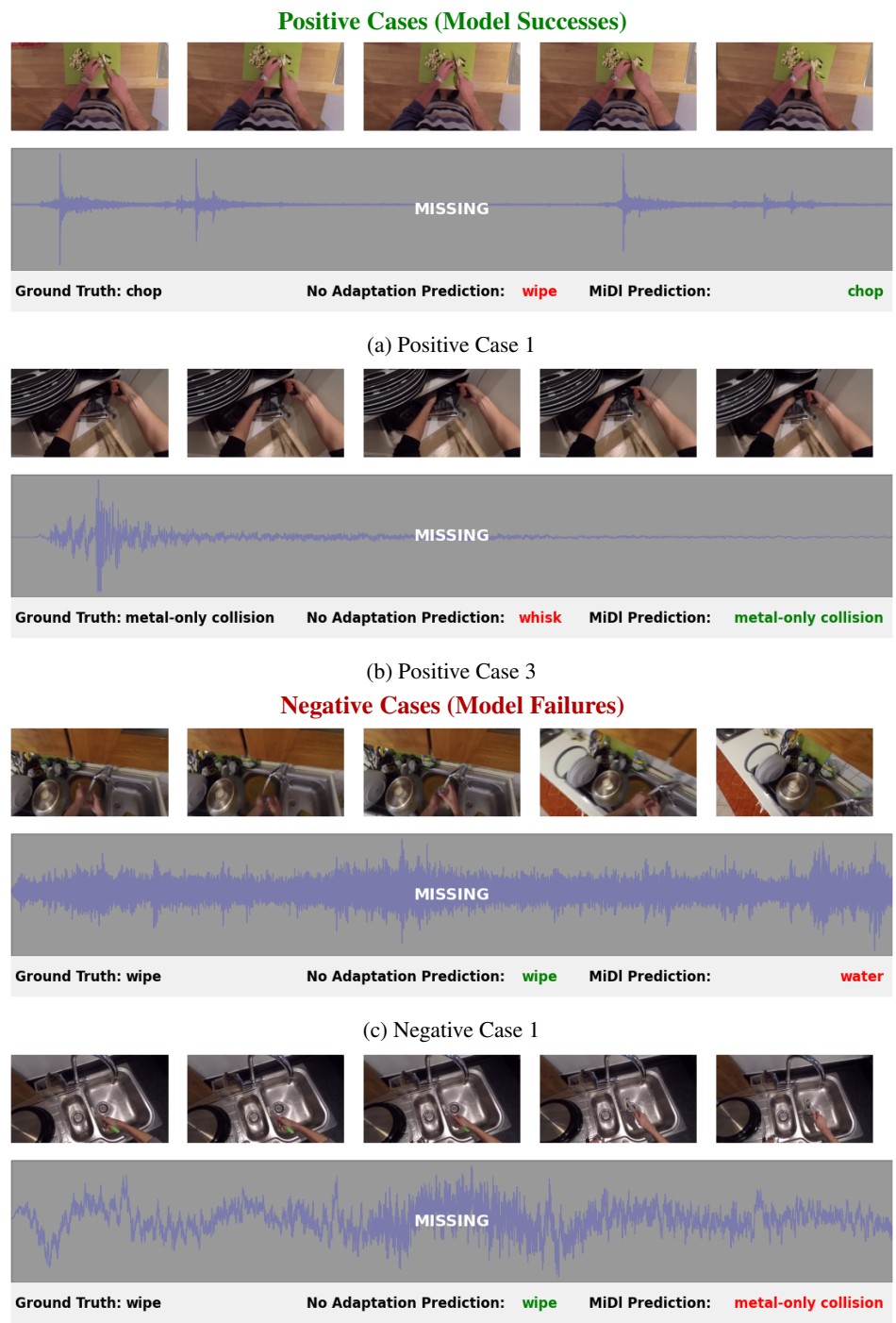

Figure 4: **Qualitative analysis of MiDl's adaptation performance on Epic-Sounds .** The top two subfigures highlight **positive cases** where MiDl successfully adapts to predict the correct label (marked in green). Conversely, the bottom two subfigures illustrate **negative cases** (marked in red) where adaptation introduces errors.

## D  QUALITATIVE RESULTS

Figure 3 presents a qualitative analysis of our method's test-time adaptation for audiovisual models on Epic-Kitchens. The top subfigures (Positive Cases) highlight successful adaptations, where the

model accurately compensates for the missing modality to predict the correct label. For example, in Positive Case 1, the model initially predicts "cupboard" without adaptation but successfully adapts to the correct label "chair" by leveraging the sound cues. This demonstrates how MiDl effectively uses auditory signals to distinguish the distinct sound of a chair. Similarly, in Positive Case 3, the model transitions from an incorrect prediction of "wash" to the correct action "mix."

In contrast, the bottom subfigures (Negative Cases) illustrate instances where adaptation introduces errors. For instance, in Negative Case 1, the model changes its correct prediction of "turn-on" (without adaptation) to an incorrect "wash," and in Negative Case 2, the method erroneously adapts from "open" (correct without adaptation) to "close."

Likewise, Figure 4 showcases the qualitative performance of our test-time adaptation method in scenarios where the audio modality is missing, using Epic-Sounds. The top subfigures (Positive Cases) demonstrate successful adaptations that align the predictions with ground truth despite the missing audio. For example, in Positive Case 1, the model refines its initial prediction from "wipe" (without adaptation) to the correct label "chop," effectively utilizing visual cues. Similarly, in Positive Case 3, the prediction is adapted from "whisk" to the correct "metal-only collision," showcasing MiDl's capacity to mitigate the absence of sound.

However, the bottom subfigures (Negative Cases) reveal errors introduced by adaptation. In Negative Case 1, the model changes a correct "wipe" prediction to an incorrect "water," potentially due to the visually confusing presence of a sink without audio context. In Negative Case 2, the method misclassifies "wipe" as "metal-only collision," likely influenced by visible metallic objects, making it a plausible yet incorrect adaptation.

Overall, Figures 3 and 4 illustrate both the strengths and limitations of our test-time adaptation method. While MiDl frequently improves prediction accuracy, as evidenced by our quantitative results, it occasionally induces misclassifications. To provide a balanced perspective, we included an equal number of success and failure cases. These failure cases offer valuable insights, paving the way for further refinements and future research.

# E    ADDITIONAL RESULTS ON EGO4D

We evaluated MiDL's performance on the Ego4D-AR dataset Ramazanova et al. (2024), derived from Ego4D (Grauman et. al., 2022), to assess its generalizability and robustness under varying levels of missing audio. Ego4D-AR encompasses diverse daily activities and naturally includes instances of missing audio. As shown in Table 12, MiDL consistently surpasses baseline methods across 50%, and 75% missing audio rates .

MiDL demonstrates significant advantages, achieving 23.41% accuracy at a 75% missing rate, compared to 21.46%, 15.92%, and 22.06% for Baseline, TENT, and SHOT, respectively. These results highlight MiDL's ability to effectively handle missing modality scenarios, adapting to challenging conditions where conventional approaches struggle.

By offering greater resilience to incomplete or noisy data, MiDL establishes itself as a robust solution for multimodal learning in egocentric video applications. These findings underscore its potential to advance state-of-the-art methods, particularly in real-world settings characterized by data sparsity or inconsistency.

Table 12: **MiDL Performance on Ego4D-AR with Missing Audio.** Performance comparison at various missing rates of audio $(1 - p_{AV})$.

| Model — $1 - p_{AV}$ (%) | Ego4D-AR (%) | | |
|---|---|---|---|
| | 50 | 75 | 100 |
| Baseline | 26.21 | 21.46 | 16.58 |
| TENT | 23.29 | 15.92 | 9.25 |
| SHOT | 26.56 | 22.06 | **18.29** |
| MIDL (ours) | **27.13** | **23.41** | 16.58 |

## CONTRIBUTION STATEMENT

We highlight the contribution of each author in this project. Merey Ramazanova coordinated and refined the project's focus on missing modality challenges in egocentric videos. She led the implementation and experimental design, including MiDl, baseline models, all multimodal setups, and ablation studies. Additionally, she managed data preparation across all datasets and co-designed the evaluation protocols. She also proposed the Out-of-Domain Adaptation setup using Ego4D.

Alejandro Pardo refined the code for large-scale experimentation, assisted in the baseline implementation, conducted the experiments, and explored various ablations and parameters for each method. He played a significant role in writing and presenting the work's experimental section and the qualitative results.

Bernard Ghanem guided the project from the beginning, helped shape the initial ideas, and significantly contributed to the design of the experiments and the project's overall development. He also proofread and polished the manuscript.

Motasem Alfarra initiated the project by formulating the missing modality challenge as a test-time adaptation problem, proposed the MiDl objective function, co-designed the evaluation protocols, led the writing efforts, and advised the implementation of MiDl along with the other baselines.

