# OpenReview forum: "Test-Time Adaptation for Combating Missing Modalities in Egocentric Videos"
_ICLR.cc/2025/Conference — ICLR 2025 Poster_

### Official Review · Reviewer_wPGf · 2024-11-03

**Soundness:** 3
**Presentation:** 3
**Contribution:** 3
**Rating:** 6
**Confidence:** 4

**Summary:**

In this work, the authors focus on an important task which is test time adaptation for egocentric video action recognition under missing modality. The authors validate existing work of TTA on this new task and propose a new method MiD1 to enhance the robustness of the learned features. The performance of the proposed method is evaluated on the EpicKitchen sound and video dataset

**Strengths:**

1. Missing modality issue is important for test time adaptation of ego centric action recognition. This task will contribute to the community.

2. Method section is clearly written and easy to follow.

3. Compared with the baseline, the proposed approach show good performance on this new task.

**Weaknesses:**

1. Lack of the comparison with other approaches specifically targeted at missing modality issue.

a. Dai, Y., Chen, H., Du, J., Wang, R., Chen, S., Wang, H., & Lee, C. H. (2024). A Study of Dropout-Induced Modality Bias on Robustness to Missing Video Frames for Audio-Visual Speech Recognition. In Proceedings of the IEEE/CVF Conference on Computer Vision and Pattern Recognition (pp. 27445-27455).

b. Lee, H. C., Lin, C. Y., Hsu, P. C., & Hsu, W. H. (2019, May). Audio feature generation for missing modality problem in video action recognition. In ICASSP 2019-2019 IEEE International Conference on Acoustics, Speech and Signal Processing (ICASSP) (pp. 3956-3960). IEEE.

c. Wang, H., Chen, Y., Ma, C., Avery, J., Hull, L., & Carneiro, G. (2023). Multi-modal learning with missing modality via shared-specific feature modelling. In Proceedings of the IEEE/CVF Conference on Computer Vision and Pattern Recognition (pp. 15878-15887).


2. The authors are suggested to enlarge the benchmarks. I aggree that this task is an important task, however the experiments are limited in this paper which will be harmful to its soundness. The authors could enrich the benchmark using more existing TTA approaches, e.g., d,e,f, and g, and try to provide an analysis on the performance on different cluster of approaches. Missing modality works can also serve as good baselines to enrich the benchmark.

d. Chen, D., Wang, D., Darrell, T., & Ebrahimi, S. (2022). Contrastive test-time adaptation. In Proceedings of the IEEE/CVF Conference on Computer Vision and Pattern Recognition (pp. 295-305).

e. Wang, D., Shelhamer, E., Liu, S., Olshausen, B., & Darrell, T. (2020). Tent: Fully test-time adaptation by entropy minimization. arXiv preprint arXiv:2006.10726.

f. Yuan, L., Xie, B., & Li, S. (2023). Robust test-time adaptation in dynamic scenarios. In Proceedings of the IEEE/CVF Conference on Computer Vision and Pattern Recognition (pp. 15922-15932).

g. Niu, S., Wu, J., Zhang, Y., Chen, Y., Zheng, S., Zhao, P., & Tan, M. (2022, June). Efficient test-time model adaptation without forgetting. In International conference on machine learning (pp. 16888-16905). PMLR.

3. No qualitative reustls. The authors are suggested to provide some qualitative results when comparing their approach with the baseline approach. Some failure case analysis will be helpful.

4. The performance of the proposed approach is only verified on EPIC Kitchen, generaliyability to other dataset can be an issue.

5. TSNE visualization on the latent space will be helpful to see how the proposed supervision help during the feature learning procedure. The authors could visualize the changes for different epoches compared with its baseline.

**Questions:**

1. Could the authors include comparisons with other approaches specifically addressing the missing modality issue, such as those proposed by Dai et al. (2024), Lee et al. (2019), and Wang et al. (2023)?

2. Given the importance of this task, would the authors consider expanding the benchmark by including more test-time adaptation (TTA) approaches, such as Chen et al. (2022), Wang et al. (2020), Yuan et al. (2023), and Niu et al. (2022), and analyzing performance across different clusters of approaches? Including missing modality approaches as baselines may also strengthen the benchmark.

3. Could the authors provide qualitative results comparing their approach to the baseline, along with a failure case analysis to offer insights into scenarios where the method may fall short?

4. Has the generalizability of the proposed approach been tested on datasets beyond EPIC Kitchen? If not, would the authors consider verifying the performance on additional datasets?

5. Could the authors use TSNE visualization on the latent space to illustrate how the proposed supervision affects feature learning? Specifically, visualizing changes over different epochs in comparison to the baseline might provide additional insights.

---

> ### Author Response · Authors · 2024-11-21
> **Responses to Reviewer wPGf's Comments and Suggestions**
>
> **On comparison with related works addressing the missing modality problem**
>
> We thank the reviewer for pointing out these related works. We would like to clarify that the proposed methods primarily address the missing modality problem during training. For example, Dai et al. investigate a strategy of randomly dropping video frames during training to improve the robustness of a multimodal system. Similarly, Lee et al. propose a method to train a network capable of generating audio features to handle missing modalities. Wang et al. focus on a multimodal learning approach that models shared and specific features for classification and segmentation tasks.
> In contrast, our work formulates the missing modality problem as a test-time adaptation challenge, a novel perspective that assumes no access to the training process or labels and instead addresses the problem entirely at test time. This distinction fundamentally differentiates our approach from the works cited, as our focus is on adapting trained models dynamically to optimize performance in the face of missing modalities.  We have added these references in the main manuscript.
>
>
> As part of this framing, we compare MiDl against existing test-time adaptation methods, which are more aligned with the assumptions and constraints of our setup. Nonetheless, we appreciate the reviewer’s suggestion and will ensure these works are acknowledged in the related work section, highlighting the distinctions between training-time and test-time approaches to the missing modality problem.
>
> ---
>
>
> **On enriching the benchmark with additional TTA methods**
>
>
> We thank the reviewer for their valuable suggestions regarding enriching the benchmark with additional TTA methods. Our work already compares MiDl to several commonly used TTA methods to validate its effectiveness, including Tent (Wang et al.) and ETA (Niu et al.), which are explicitly mentioned in the manuscript. Results for these methods are presented in Tables 1, 3, and 7, showcasing their performance under different scenarios and comparing them to MiDl.
>
>
> The primary goal of our work is to redefine the missing modality problem as a test-time adaptation challenge, introducing a novel approach where pretrained models are adapted at test time to optimize performance in the face of missing modalities. We conduct extensive experiments, including ablations across various scenarios such as different backbones, pretraining strategies, and modality setups, to demonstrate MiDl’s effectiveness.
>
>
> While we appreciate the suggestion to include additional methods like Contrastive TTA (Chen et al.) and Robust TTA in dynamic scenarios (Yuan et al.), we emphasize that our current comparisons and analyses already provide a comprehensive evaluation of MiDl’s performance. Future work could further expand on these comparisons to include additional methods for broader validation.
>
>
> ---
>
>
> **On providing qualitative results and failure case analysis**
>
> Thank you very much for this valuable suggestion. We are currently preparing qualitative examples comparing our approach with the baseline. These examples, along with an analysis of failure cases, will be included in the supplementary material in the revised submission.
>
>
> ---
>
>
> **On generalizability beyond Epic-Kitchens**
>
> Thank you for pointing this out. As mentioned in Section 5.1, we validate our approach on two distinct datasets: Epic-Sounds (Huh, et. al., 2023) and Epic-Kitchens (Damen, et. al, 2020). To align with the experimental setup of prior work, we assume different missing modalities for each dataset, with video missing in Epic-Kitchens and audio missing in Epic-Sounds. This demonstrates the adaptability of our method to varying modality configurations.
>
>
> ---
>
>
> **On TSNE visualization of latent space**
>
> We thank the reviewer for this insightful suggestion and their interest in understanding the effects of our method. While TSNE visualization is commonly used to illustrate feature learning and the clustering behavior of learned representations, we would like to emphasize that MiDl is not a feature learning approach in the traditional sense. Instead, it focuses exclusively on adapting pretrained models at test time by updating only the parameters of the normalization layers to handle missing modalities dynamically.
>
> This design choice means that MiDl does not aim to significantly alter the learned feature space but rather adjusts the model's predictions to maintain robustness under test-time conditions. Consequently, the use of TSNE to visualize changes across epochs may not be directly relevant to evaluating MiDl’s effectiveness.
>
> If the reviewer has specific aspects of the latent space or adaptation process they would like to see explored, we would be happy to incorporate such analyses to further enhance the interpretability of our method.

---

> > ### Author Response · Authors · 2024-11-24
> > **Qualitative Results Update**
> >
> > Dear Reviewer wPGf,
> >
> > In response to your request for qualitative results, we have added Figures 3 and 4 in Section D of the appendix (please refer to the updated version of the PDF). These figures compare our approach with the base model and include a failure case analysis. This addition provides valuable insights into the strengths of our method and identifies scenarios where it may face limitations, directly addressing your feedback.
> > We appreciate your thoughtful suggestion, as incorporating these results has enhanced both the presentation and the overall impact of our paper’s findings.

---

> > ### Comment · Reviewer_wPGf · 2024-11-25
> > **Response to the rebuttal**
> >
> > Dear Authors,
> >
> > I am not satisfied with your answer regarding the generalizability of your approach beyond EPIC-Kitchen.
> >
> > The two datasets you mentioned are all from EPIC-Kitchen but in different modalities.
> >
> > Some other datasets can be also leveraged to conduct the experiments and validate the generalizability of your proposed method, e.g., Ego4D dataset (or other dataset) [1].
> >
> > [1] Grauman, K., Westbury, A., Byrne, E., Chavis, Z., Furnari, A., Girdhar, R., ... & Malik, J. (2022). Ego4d: Around the world in 3,000 hours of egocentric video. In Proceedings of the IEEE/CVF Conference on Computer Vision and Pattern Recognition (pp. 18995-19012).
> >
> > I think the task proposed by the authors are interesting, but the experiments are only conducted on EPIC-Kitchen which may limits the contribution.
> >
> > Thereby I will keep my score as 5 based on the current response.

---

> > > ### Author Response · Authors · 2024-11-27
> > > **Updates on Dataset Diversity**
> > >
> > > Thank you for your thoughtful feedback and for engaging deeply with our work. We truly appreciate your efforts to ensure that the contributions of our approach are robust and well-supported, and we welcome the opportunity to further clarify our choices and provide additional evidence of MiDl's generalizability.
> > > As requested by the reviewer, we conducted experiments with MiDl on Ego4D dataset. Although Ego4D does not currently provide an official action recognition benchmark, we adopt the approach of Ramazanova et al. (2023) and use their Ego4D-AR dataset.  In our evaluation, **MiDl demonstrates consistent improvements over baseline methods while dealing with missing audio in Ego4D-AR**.  For instance, at a 75% missing rate, MiDl achieves 2% performance gain over the baseline achieving an accuracy of 23.4%, outperforming the baseline (21.4%), TENT (15.9%), and SHOT (22.1%). **These findings are detailed in Table 12 in the PDF**. Notably, as Ego4D inherently features instances of missing audio, we conducted evaluations at 50%, 75%, and 100% missing rates.
> > >
> > >
> > > | 1-p_{AV} (%) | 50    | 75    | 100   |
> > > |--------------|--------|-------|-------|
> > > | BASELINE     | 26.2% | 21.4%| 16.6%|
> > > | TENT         | 23.3% | 15.9%| 9.3% |
> > > | SHOT         | 26.6% | 22.1%| **18.3%**|
> > > | MIDL (ours)  | **27.1%** | **23.4%**| 16.6%|
> > >
> > >
> > > ---
> > >
> > >
> > > We would also like to emphasize that EPIC-Kitchens (Damen et al., 2020) and EPIC-Sounds (Huh et al., 2023) are two distinct datasets with no overlap in their set of classes or recognition tasks. While both datasets originate from the same underlying collection of long-form videos recorded in kitchen environments, **each dataset comprises a distinct set of trimmed clips**. Moreover, their **annotations and tasks are different**: EPIC-Kitchens focuses on action recognition, while EPIC-Sounds is designed for audio/sound classification. That said, we acknowledge that both datasets are centered on kitchen activities, and Ego4D (Grauman et al., 2022) encompasses a broader range of daily activities.
> > >
> > >
> > > We hope this additional experiments and clarification  sufficiently address your concerns regarding generalizability. Thank you again for your constructive feedback, which has motivated us to strengthen our work further.

---

> > > > ### Comment · Reviewer_wPGf · 2024-11-28
> > > > **Response**
> > > >
> > > > Dear authors,
> > > >
> > > >
> > > > Thank you for your detailed response. My concern is well solved by the latest reponse from the authors and I will increase my rating to 6.
> > > >
> > > > Best,
> > > >
> > > > your reviewer.

---

> > > > > ### Author Response · Authors · 2024-12-02
> > > > > **Response**
> > > > >
> > > > > Thank you for taking the time to thoughtfully review our rebuttal and for reconsidering your scores. We greatly appreciate your constructive feedback and recognition of our work's contributions. Your insights have been invaluable in improving the quality of our work.

---

### Official Review · Reviewer_M6ux · 2024-11-03

**Soundness:** 3
**Presentation:** 4
**Contribution:** 3
**Rating:** 6
**Confidence:** 3

**Summary:**

This paper presents a novel approach to handling missing modalities in multimodal learning using test-time adaptation. The method, MiDl, shows promising results across the EPIC kitchen and EPIC sounds datasets, and the method is motivated by the theoretical intuition of minimizing the mutual information between the predicted and available modality. The authors also provide some interesting analysis of the model through long-term adaptation, out-of-distribution warmup, and various ablation experiments.

This review follows the sections of the paper.

**Strengths:**

Introduction:
1. The second and third paragraphs effectively identify the gap in the literature and provide a robust overview of the proposed solution.

Related Works:

2. Related works is concise and relevant

Missing Modality as Test Time Adaptation:

3. This section is well-written and easily comprehensible.

Experiments:

4. It is commendable that the experiments were repeated 5 times with reported standard deviations in Table 11; the results appear experimentally robust and convincing.
5. The Ego4d Warmup result on 100% missing rate in the original distribution is an interesting finding with potentially strong applications.

Analysis on MiDL:

6. The Components of MiDl section is strong, and the ablations are empirically sound.

**Weaknesses:**

Introduction:
1. The motivation presented in the first paragraph is weak. For instance, in Line 40, could you provide an example or application where inference must occur on a redacted modality? I can think of the application of blurring faces in images or deleting private details in medical records, but it's unclear when only one modality would be completely removed for privacy reasons while the other remains intact for the same data instance. Additionally, the relevance of using cheaper modalities to missing modalities is not apparent (line 40). It would be particularly convincing if the motivation aligned with the tested dataset. For example, if using Epic Kitchens, perhaps a scenario involving a humanoid with a malfunctioning audio sensor, or smart glasses with an obscured camera due to steam or food spillage could be considered.
2. Contribution (3) appears to be describing MiDl, which is already covered in contribution (2). I would recommend reassessing what could constitute a third distinct contribution from your work.
3.Figure 1 requires improvement. The concept of a "potential performance trajectory" needs clarification - is this your hypothesis? This graphic would be more persuasive if it depicted your actual no-adaptation baseline and your TTA method. The purpose of the black line in the middle of the graph is unclear.

Proposed Solution:

4. The notation in eq (1) lacks precision. It is not evident that f(x;m) and m are random variables. The output of f is a distribution. Are you considering this as a random variable, with the value being the indices and the probability of those values the logits? Consider introducing a random variable Y ~ f(x;m). Also, consider using capital letter notation (e.g. "M") for the random variables. Furthermore, how can you evaluate the KL if x ~ S only has the modality m, not AV? Later in this section, it becomes apparent that you only update the model on complete instances. This limitation/assumption should be made clearer in the introduction or Takeaways subsection. This method would only be applicable for testing data that includes some multimodal instances.
5. At last line of page 4, is $x_t$ a single sample? Do you mean samples $x_0 \dots x_t$?

Experiments:

6. Additional details could enhance the reproducibility of this work. Was any hyperparameter tuning conducted for MiDl? Section B.1 mentions the recommended hyperparameters for the baseline but doesn't specify how they were determined for MiDl. Moreover, what were the proportions of the train/val/test split?
7. In section 5.3 LTA: You allow the model to retrain on some of the unlabeled training data. Why not gather $S_{in}$ from the validation set? In this setting, is MiDl trained on $D \ S_{in}$? Or is $S_{in}$ still included in the labeled dataset before test time and then used without labels during test time?
8. Can this method be applied to instances where either modality is missing, e.g., P = {.3,.3,.3}? It would be great to see results for experiment with such a ratio. Currently, it may be the case that the model learns to leverage ONLY the modality that is consistently present in both the complete modality test case and the missing modality test case. In this scenario, would a given unimodal model for the always-present modality perform optimally? Table 1 could be improved by clarifying what is meant by "Unimodal" and why it is only present at the 50% missing rate. For Epic Sounds, is the unimodal the always-present modality (video)?

Analysis on MiDL:

9. Both architecture choices are transformer-based. It would have been more convincing to see a greater diversity of architectures (such as a convolution backbone). Instead of presenting different missing rates as columns in Table 3, it would have been preferable to see different architectures/methods as the columns with a fixed missing rate (perhaps 50%).
10. Given that the main motivation was to avoid retraining an existing method on a large dataset to perform missing modality adaptation, the results would have been more convincing if the authors had either used an existing model+dataset and just performed adaptation, as opposed to training from scratch an existing method. Alternatively, they could have tested with a very large dataset that was computationally expensive. The omnivore pretraining test is good. Did you train the model from scratch on your dataset or use an existing model and apply MiDl?
11. In Table 6, shouldn't 55.2 in the Dl column be bolded?
12. I thought the motivation was that retraining on the train set is computationally expensive, and TTA will prevent that? It's good that you acknowledge the computational requirements of MiDl, but then in the abstract, you shouldn't state: "Current methods, while effective, often necessitate retraining the model... making them computationally intensive." Alternatively, compare your inference computation here with the amount of computation required to retrain training data (to get sufficient performance).

**Questions:**

While the technical aspects and experimental results are generally strong, there are areas for improvement in the motivation, clarity of presentation, and some experimental details.

I presented many questions and suggestions in the weaknesses suggestions. In particular, I would suggest the authors focus on the concerns about the motivation and the experiments aligning with that motivation. My comments regarding notation and small fixes are merely suggestions.

---

> ### Author Response · Authors · 2024-11-21
> **Responses to Reviewer M6ux's Comments and Suggestions**
>
> **Introduction**
>
> **Motivation and Missing Modality Examples**
> We thank the reviewer for their thoughtful feedback and detailed suggestions. The necessity of addressing this problem can indeed be observed in practical scenarios such as the large-scale egocentric data collection in the Ego4D dataset (Grauman et al., 2022). In this dataset, while RGB video is consistently available, audio is absent in approximately 30% of the data due to privacy regulations in specific locations.
> Additionally, the reviewer’s suggestion about device failure or sensor deactivation is a valid and practical scenario. We also appreciate the reviewer’s point about the relevance of using cheaper modalities. This topic has been explored in recent literature (Grauman et al., 2023), where researchers focus on scenarios where models can infer information using fewer, less costly modalities to reduce energy consumption.
>
>
> ---
>
> **Contributions and Figures**
>
> **Contributions and Figure 1 Improvements**
> We thank the reviewer for their suggestions regarding the figures and contributions. We have updated Figure 1 and the contributions statement accordingly in the manuscript. To clarify, the "potential performance trajectory" in the original figure was intended to represent the desired outcome of a successful adaptation strategy, serving as a conceptual illustration rather than being based on actual data. However, we agree that using our method's results instead makes the figure more impactful and better supports the message. We replace this plot with MiDl results on the Epic-Kitchens dataset.
>
>
> ---
>
> **Proposed Solution**
>
> **Notation and KL Divergence Clarifications**
> We thank the reviewer for raising these points.
>
> 1. **Regarding Notation**: Indeed, $m$ is a discrete random variable sampled from $\{A, V, AV\}$, defined by probabilities $\{p_A, p_V, p_{AV}\}$, a property of the stream $\mathcal{S}$. For instance, $p_{AV} = 1$ implies $\mathcal{S}$ always reveals complete modalities. Sampling $x \sim \mathcal{S}$ equips $x$ with $m = M$. We are happy to revise the notation further if necessary.
>
> 2. **Regarding KL Divergence**: When $x$ is modal-complete, we define the KL divergence expectation over $m$ with $p_A = p_V = p_{AV} = \frac{1}{3}$.
>
> 3. **Regarding Updates**: As outlined in the interaction (gray box) in Section 4 (lines 262-263), MiDl adapts the model on the modal-complete data points while predicting without adaptation on other samples. Thus, MiDl produces predictions for every single data point regardless of its modality. As noted in lines 265-265, our work focuses on multimodal settings, and we assume $p_{AV} \neq 0$.  We have modified the Takeaways subsection of Section 4 to more explicitly highlight this assumption.
>
> 4. **Clarification on x_t**: We apologize for any confusion regarding this notation. As clarified in lines 179–181, x_t refers to a sample or batch presented to the model at time step t. This does not imply all samples accumulated up to step t (i.e., x_0, x_1, \ldots, x_t​); rather, it strictly refers to the data arriving at time step t alone.
>
> ---
>
> **Experiments**
>
> **Reproducibility Enhancements**
> We appreciate the reviewer’s request for additional details.
>
> 1. **Hyperparameter Tuning**: The implementation details for MiDl, including the selected hyperparameters, are provided in Section B.1. These hyperparameters were determined through a grid search to identify the optimal settings for the task. We will ensure that this clarification is made explicit in the manuscript. Additionally, our code release will further facilitate the reproducibility of these results.
>
> 2. **Dataset Splits**: We adhered to the official train/val/test splits provided for the Epic-Kitchens and Epic-Sounds datasets. The approximate ratios for these splits are 75% for training, 10% for validation, and 15% for testing. We will revise the manuscript to explicitly state these proportions to avoid any ambiguity.
>
> **Clarification on LTA Setup**
>
> We thank the reviewer for this thoughtful observation. To clarify, as outlined in Section 5.3, we reserve a subset of the training data for the Long-Term Adaptation (LTA) stage. The model observes labeled data from S_{in} prior to test time (during training), but we do not use any labels—whether from S_{in} or elsewhere—during the adaptation phase. This design simulates a practical scenario where a portion of training data can be stored and utilized for adaptation at test time without relying on labels.
> We do not use validation or test data for LTA because our assumption is that data arrives as a stream at test time, requiring immediate predictions. While our current setup reflects this realistic streaming assumption, in practical scenarios, one could envision access to test data in advance, allowing for storage of unlabeled data for long-term adaptation. This flexibility could further enhance the applicability of MiDl in various deployment settings.
>
> ---

---

> ### Author Response · Authors · 2024-11-21
> **Responses to Reviewer M6ux's Comments and Suggestions**
>
> **Exploring Missing Modality Ratios**
> We thank the reviewer for these insightful observations. This scenario is indeed valuable to explore. As noted in Appendix B.4 and Table 9, we report results for the mixed missing modality setup, where either modality may be absent.  These results demonstrate that MiDl consistently outperforms the baseline under all tested conditions, including scenarios with mixed modality availability.
>
>
> **Unimodal Clarifications**
> We appreciate the reviewer’s comments on clarifying the meaning of "Unimodal." In our manuscript, unimodal refers to a model that uses only the always-present modality (e.g., video for Epic-Sounds and audio for Epic-Kitchens). We apologize for the confusion caused by the presentation in Table 1, where the unimodal result appears only at the 50% missing rate. To clarify, the unimodal results are constant across all missing rates, as the model relies solely on the non-missing modality, which remains unaffected by the missing rate of the other modality. To avoid redundancy, we initially reported the unimodal result once in the middle of the table, but we acknowledge that this presentation may have caused confusion. We have revised the manuscript to explicitly show the unimodal results across all missing rates for clarity.
>
> ---
>
> **Architecture Diversity**
>
> We thank the reviewer for this insightful comment.
> We would like to clarify that we do present results with different architectures and models. Specifically, we report results for self-attention-based models in Section 6.1 and Omnivore in Section 6.2. While we agree that further exploration with more diverse setups (e.g., convolutional backbones) could be valuable, our focus was on evaluating state-of-the-art and widely-used architectures, which are predominantly transformer-based.
> We appreciate the suggestion to reorganize Table 3 to present results with different architectures under a fixed missing rate. While our current presentation emphasizes performance across varying missing rates, we recognize that including architecture-level comparisons could provide complementary insights. We are committed to releasing the code, which we hope will enable further exploration of this problem from an architectural perspective.
>
> ---
>
> **Computational Efficiency**
>
> **TTA vs. Retraining**
> We apologize for any confusion. As mentioned in Section 4 (Lines 270–275), we formulate the missing modality challenge within the test-time adaptation scenario. In this framework, we make no assumptions about the training process. Instead of retraining the network, we adapt it at test time by updating only a small subset of parameters.
>
>
> **Omnivore experiment**
> Our approach is designed to work with existing pretrained models, as demonstrated in our experiments, including the Omnivore pretraining test. This emphasizes the practicality of MiDl, as it eliminates the need for retraining on large datasets, aligning with our primary motivation.
>
> ---
>
> **Minor Points**
>
> **Bolded Numbers Table 6**
> We thank the reviewer for catching this oversight. The correct value (55.2 in the "Dl" column) has been bolded in the updated manuscript.

---

> ### Comment · Reviewer_M6ux · 2024-11-27
>
> The authors have done a fairly good job in their rebuttal, addressing my suggestions and making reasonable changes to the work. However, I remain unconvinced regarding the privacy-related motivation and the suggested computational benefits.
>
> While Ego4D redacted modalities due to privacy concerns, this was done for training purposes and model construction. It's unclear why a modality would be missing for privacy reasons when deploying the model in a real-world setting. This may be a misunderstanding or misphrasing in the introduction.
>
> Regarding computational efficiency, I think the authors misunderstood my original stated weakness. The work mentions that "MiDl is 5x more expensive" during testing. This raises the question: if the test set is more than one-fifth the size of the training set, wouldn't retraining a different model be faster than adapting with MiDl? I believe more analysis is needed to convincingly demonstrate that MiDl TTA is computationally superior to retraining.
>
> Nevertheless, this is the first work I've encountered that explicitly explores missing modality TTA. Although there's room for expansion in terms of the number of modalities used, datasets employed, and baselines compared against, I believe the work provides a modestly sufficient contribution for ICLR. The exploration of various related and interesting aspects of this problem, such as different missing rates and LTA, is also noteworthy.
>
> Consequently, I am revising my score to marginally above acceptance.

---

> > ### Author Response · Authors · 2024-12-02
> >
> > Thank you for taking the time to thoughtfully review our rebuttal and for reconsidering your scores. We greatly appreciate your constructive feedback and recognition of our work's contributions. Your insights have been invaluable in improving the quality of our work.

---

### Official Review · Reviewer_CsYs · 2024-11-04

**Soundness:** 3
**Presentation:** 3
**Contribution:** 3
**Rating:** 6
**Confidence:** 4

**Summary:**

This paper tackles on missing modalities in egocentric videos without the need to retrain models by formulating this challenge as a test-time adaptation task. The authors proposed MiDl which minimizes the mutual information between prediction and the modality, with the incorporation of self-distillation to maintain performance when all modalities are available. The author benchmarked several methods under such problem formulation, demonstrating a descent performance when part of the modality are missing in two egocentric video datasets.

**Strengths:**

Overall, the paper is interesting and easy to follow.
- The formulation of the test-time adaptation for tackling missing modality without the need for model retraining is indeed novel and can be foreseen to be applied to various applications which contains multi-modal information.
- Although the method itself is not complex and consists of components already used for various tasks for multi-modal learning and egocentric video analysis, they are leveraged in MiDl with strong motivation which are intuitive and reasonable. The extended discussion also offers a deeper understanding over the formulation of MiDl. This method could prove to be a good starting point for subsequent discussions in the research community.
- The authors also provide a comprehensive analysis over the performance of MiDl in the formulated task, and also benchmarked previous methods such as SHOT, TENT under the same setting, which also provides a further insight into the challenges and possible methods to further tackle the task.

In general, this paper is relatively well presented with a simple yet highly motivated method for an interesting formulation of a realistic challenge.

**Weaknesses:**

There are a few minor concerns remaining in the paper, mainly on the clarity and possible extension in discussion of the proposed method. I would like the authors to consider the following concerns if possible:
1. On Page 4, Line 197-198, the author states that "$f_\theta$ should retain high performance in predicting data with complete modality, which is generally satisfied for $f_{\theta_0}$". Does this imply that the non-adapted pretrained model must be pretrained with all modalities available? What if the pretrained model is only trained with a single modality (e.g., only with visual information without the audio information which is rather common in video models)?
2. It is observed that there is a large drop when $1-p_{AV}=100$, where none of the data contain both modalities. What would be a possible approach to mitigate this drop in performance. It is observed that the drop for MiDl is significantly more severe than that of SHOT.
3. The current method only touches upon the case for two modalities (audio and video), is it expandable towards more modalities. Also, are there limitations for the possible types of modalities or it can be any modalities as long as they are obtained from the same set of data?
4. The experiments are performed for each dataset with a drop in the primary modality, what would be the result if the secondary modality is dropped with the same probability?
5. Lastly, the code is currently NOT available, which means that the reproducibility of the result is not verified.

**Questions:**

Please refer to the Weakenesses section. I highly encourage the authors to directly include the code for the verification of reproducibility.

---

> ### Author Response · Authors · 2024-11-21
> **Responses to Reviewer CsYs's Comments and Suggestions**
>
> **On multimodal pretraining requirements**
>
> Thank you for raising this question. We assume that the model accepts multimodal inputs at test time and has been trained using multimodal data to ensure compatibility with the test-time scenario. If a unimodal model were used at test time, it would lack the capability to leverage the full set of modalities present in the multimodal inputs, thereby limiting its performance and effectiveness in such scenarios.
> To address the scenario you mentioned, where a unimodal model (e.g., a video model) is used for multimodal data, a finetuning stage would typically be required. For instance, the audio backbone could be initialized with the weights from video pretraining and then finetuned on audiovisual data. However, in our method, we do not perform any finetuning. Instead, we assume that the models have already been trained on multimodal data, which aligns with the scope and assumptions of our approach.
>
> ---
>
>
> **On mitigating performance drops at 100% missing modality**
>
> Thank you for your insightful comments. Adapting a model to a fully unimodal stream (100% missing rate) at test time is indeed a challenging scenario, particularly without labeled data. In this extreme case, MiDl neither degrades nor improves baseline performance, maintaining the integrity of the original multimodal model.
> While methods like SHOT provide slight improvements over the baseline under a 100% missing ratio, they exhibit significantly lower performance compared to MiDl when some multimodal samples are available. MiDl is designed with the assumption that some presence of complete modalities at test time is necessary for effective adaptation, which aligns with the typical expectations for multimodal models.
> We view it as a strength of MiDl that it avoids degrading the original model’s performance in this extreme case, rather than a limitation. Moreover, we highlight that MiDl demonstrates a significant performance boost during long-term adaptation, even if the test stream becomes unimodal over time (see Table 2 for detailed results).
>
>
> ---
>
>
> **On expandability to more modalities**
>
> We thank the reviewer for this insightful comment. MiDl does not impose any inherent limitations on the number of modalities; the scalability depends on the capabilities of the base model used. In Section 3, we formulated our approach m ∈ {A, V, AV} to align with our experiments on audiovisual egocentric datasets. However, the underlying problem and methodology can naturally extend to any number of modalities.
> Similarly, MiDl is designed to work seamlessly with an arbitrary number of modalities. The formulations in Equation 1 and Equation 2 can be easily generalized by replacing AV with combinations of additional multimodal inputs, enabling broader applicability beyond the audiovisual setup presented in this work.
>
> ---
>
>
> **On dropping the secondary modality**
>
> We thank the reviewer for this comment. We report results for scenarios where the secondary modality is dropped in Section 6.2 and Table 4 of the main manuscript. Specifically, we present results for Epic Sounds when the video modality is dropped and for Epic Kitchens when the audio modality is dropped. These experiments demonstrate the robustness of our method across different modalities under varying missing probabilities.
>
> ---
>
>
> **On reproducibility of results**
>
> We are working on a code release, please stay tuned for future replies. We are committed to submit it before the discussion period ends.

---

> > ### Comment · Reviewer_CsYs · 2024-11-26
> > **Response to Rebuttal**
> >
> > I thank the authors for their effort in answering my doubts and concerns. I would maintain my rating at the moment. I would like to see if the code can be released shortly to check for reproducibility.

---

> > > ### Author Response · Authors · 2024-11-29
> > > **Updates on the code release**
> > >
> > > We sincerely thank the reviewers for their patience regarding the code release. We are pleased to inform you that we have updated the manuscript to include an anonymous link to the code, which you can access [here](https://anonymous.4open.science/r/midl_tta-2E36/). The repository includes a detailed README file with all the necessary details for running the code. Upon acceptance, we are fully committed to making this repository publicly available.

---

### Official Review · Reviewer_6iPc · 2024-11-06

**Soundness:** 3
**Presentation:** 2
**Contribution:** 3
**Rating:** 6
**Confidence:** 4

**Summary:**

To tackle the issue of modality missing in real-time tasks, this framework offers an online self-supervised learning method called MiDl. MiDl uses mutual information and KL divergence as loss functions to optimize the model in real time, enabling the baseline model to better handle inputs with missing modalities.

**Strengths:**

The paper redefines the modality missing problem as a test-time adaptation (TTA) issue, emphasizing the challenges of modality absence faced in online multimodal tasks. This is indeed an urgent problem that needs to be addressed for many online multimodal tasks.

The proposed MiDl method effectively enhances the baseline model's ability to handle missing modalities, serving as a solution for the modality missing problem in multimodal online tasks. This approach can act as a supplement when facing modality absence in such tasks. For instance, if modalities are functioning normally, this pipeline may not be used; however, when a modality is missing, the proposed solution can improve the baseline model's capability to handle the missing modalities. Additionally, normal inputs and prediction results can serve as supplementary information when modalities are insufficient.

The experiments presented in the paper are comprehensive, demonstrating that the
method is independent of modality selection, baseline models, and model frameworks, thereby proving the robustness of the proposed solution.

**Weaknesses:**

1. "First, the prediction of should be invariant to the modality source . Ideally, f_{\theta} should output the same prediction under both complete and incomplete modality, hence satisfying the following equality: (i)...," The underlying assumption of the approach is controversial. The task will degenerate into a modality distillation problem if this assumption holds,. Is there a more reasonable way to phrase this?
2. Implementing this method in real-world production could introduce significant computational overhead and latency. Normal models can be accelerated through techniques like compression and distillation, but this approach involves updating model weights, requiring the retention of the complete model, making it difficult to deploy directly in practice.
3. Could you include experiments demonstrating the approach's decision-making in more complex online scenarios? The experiments provided in the paper do not represent the best use case for this method; its most suitable application is in online scenarios, so experiments in these contexts would better support the results.

**Questions:**

1. If this approach faces extreme examples, such as a video showing a calm street while the audio is an explosion, will this model mislead the baseline model into the wrong direction?
2. You might consider adding extra blocks to the model, so that if updates are needed, only the added portions need to be updated. Alternatively, updating part of the model's structure could prevent the significant latency introduced by updating the entire system.

---

> ### Author Response · Authors · 2024-11-21
> **Responses to Reviewer 6iPc’s Comments and Suggestions**
>
> **On the assumption and phrasing regarding modality invariance**
>
>
> We thank the reviewer for raising this insightful point. We would like to clarify the distinction between our setup and the modality distillation problem, as they address fundamentally different challenges. In modality distillation, the primary goal is to transfer knowledge from a teacher model, which is usually trained with access to all modalities, to a smaller, typically unimodal student model. The focus is on training a student model to approximate the teacher’s performance despite having access to fewer (or weaker) modalities.
>
> In contrast, our approach assumes that the model should inherently possess the capability to make consistent and accurate predictions, irrespective of the available modality or combination of modalities. For example, whether the model observes a silent video of a bird, hears the chirping sound alone, or has access to both, it should consistently recognize that it is observing a bird. This is not a process of distillation from one model to another but rather an effort to provide a single, modality-agnostic model that learns to generalize across different modality combinations.
>
> ---
>
> **On computational overhead and latency in real-world applications**
>
>
> We thank the reviewer for highlighting this important consideration. We would like to emphasize that MiDl is agnostic to the base model, meaning it makes no assumptions about the model architecture or pre-training strategy. Consequently, our method can be applied even if the pre-trained model has already been compressed or distilled.
>
> We acknowledge the computational cost associated with test-time updates, as noted in Section 6.5. This trade-off is a common consideration for any test-time adaptation methods and should be weighed against the performance benefits in real-world deployments.
>
> ---
>
> **Extra experiments in online scenarios**
>
>
> Thank you for this valuable suggestion! We agree that addressing more complex online scenarios would be a compelling direction for further exploration. In fact, we followed the recent literature on online test-time adaptation (Alfarra et al., 2023) to define our stream setting, ensuring alignment with the current state-of-the-art. We are open to exploring additional scenarios that could showcase the applicability of our approach in even more complex online settings. Could you please elaborate on the specific scenarios or challenges you believe would better demonstrate the utility of our method? We would greatly appreciate your input.
>
>
> ---
>
> **On extreme examples with misaligned modalities**
>
>
> We appreciate the reviewer’s curiosity and for bringing up this interesting scenario. However, we would like to clarify that this situation is beyond the scope of our paper. The scenario you describe pertains to misalignment in multimodal data, which differs fundamentally from the missing modality problem that we address. In the missing modality problem, we are aware that a modality is absent, which may occur due to device malfunctions, sensor deactivation for privacy or efficiency, or similar reasons, as discussed in the introduction.
>
> In the misalignment scenario you outlined, all modalities are still present but are not semantically aligned. If the misalignment is intentional, such as injecting incorrect inputs, it may fall under the category of adversarial attacks. This is outside the focus of our work, as we concentrate on scenarios where a modality is simply missing and not replaced with deliberately crafted or noisy inputs.
>
> Even in less extreme cases of natural misalignment—such as a TV playing unrelated sounds in the background while the video reflects what a person sees—this situation involves all modalities being available and presents a different setup to the one tackled by our work. Our focus remains on the challenges and solutions specific to missing modality scenarios.
>
>
> ---
>
> **On adding blocks or partial updates**
>
>
> Thank you for this interesting suggestion. Efficiency and latency are indeed critical considerations when applying test-time adaptation methods. As mentioned in Section B.1 (Lines 735–739 of previous document, or 751-755 on updated document) and Section B.3, our approach follows the prior line of work in test-time adaptation methods by only updating the normalization layers when applying MiDl, which reduces computational overhead.
>
> Additionally, it is important to note that as we mentioned in Section 6.1, our approach is architecture-agnostic. This flexibility allows users to opt for a more lightweight architecture equipped with MiDl to tailor their specific application, thereby further addressing concerns around efficiency and latency.

---

### Author Response · Authors · 2024-11-21
**General reply**

We sincerely thank the reviewers for their insightful feedback and recognition of the strengths of our work. We are encouraged by the positive reception and appreciate the reviewers highlighting several key aspects of our contributions:




**Novelty of Problem Formulation**


We are grateful that the reviewers acknowledged our redefinition of the missing modality problem as a test-time adaptation (TTA) challenge. This novel formulation **eliminates the need for retraining** and provides a practical solution for multimodal online tasks. Reviewer 6iPc emphasized the urgency of addressing this challenge in online settings, while Reviewer CsYs noted its potential to inspire further exploration in the research community.


**Effectiveness and Generalizability of MiDl**


We appreciate the recognition of MiDl's adaptability and robust performance across different modalities, baseline models, and frameworks. Reviewer 6iPc commended MiDl as an effective solution for missing modalities, while Reviewer wPGf noted its strong performance compared to baselines. Reviewer CsYs highlighted the **comprehensive experimental evaluation**, demonstrating MiDl's applicability to various scenarios.


**Clarity and Motivated Presentation**


We are pleased that the reviewers found the manuscript to be clear and well-organized. Reviewer CsYs appreciated the **strong motivation** behind our method and its intuitive design. Reviewer wPGf also noted that the method section is easy to follow, which aligns with our goal of presenting a practical and accessible solution. Additionally, Reviewer M6ux highlighted that Section 3 ("Missing Modality as Test-Time Adaptation") is well-written and easily comprehensible.




**Comprehensive Experiments and Insights**


The **thoroughness of our experiments was highlighted by multiple reviewers**. Reviewer M6ux commended the robustness of our experimental setup, including repeated trials with standard deviation reporting, as well as the significance of findings such as the Ego4D warm-up results at a 100% missing rate. Reviewer 6iPc also pointed out that the experiments effectively demonstrate the robustness of the proposed method, as it is independent of modality selection, baseline models, and model frameworks. Reviewer CsYs also acknowledged our extensive analysis and benchmarking of prior TTA methods such as SHOT and TENT, which provide valuable insights into the formulated task.




**Relevance and Broader Impact**


We are encouraged that the **reviewers recognize the broader impact of our work** on the research community. Reviewer wPGf highlighted the importance of the missing modality issue for ego-centric action recognition, while Reviewer CsYs noted that our method offers a foundation for subsequent discussions and developments in this area.




We deeply appreciate the reviewers' constructive feedback and their acknowledgment of the strengths of our work. These insights will help us further refine our manuscript and reinforce its contribution to the field. Given the positive reception of the paper and its potential for future research, we are committed to releasing the code before the rebuttal period ends. We are actively working on it.

---

### Meta-Review · Area_Chair_Khux · 2024-12-21

**Metareview:**

The rebuttal provided clarifications about the proposed method and its analysis that were useful for assessing the paper's contribution and responded adequately to most reviewer concerns. All reviewers recommend acceptance after discussion (with four marginally above the acceptance threshold), and the ACs concur. The final version should include all reviewer comments, suggestions, and additional clarifications from the rebuttal.

**Additional Comments On Reviewer Discussion:**

NA

---

### Decision · Program_Chairs · 2025-01-22

Accept (Poster)